# Bulk-suppressed and surface-sensitive Raman scattering by transferable plasmonic membranes with irregular slot-shaped nanopores

Roman M. Wyss [1,2], Günter Kewes[1], Pietro Marabotti [1], Stefan M. Koepfli[3], Karl-Philipp Schlichting[4], Markus Parzefall [5], Eric Bonvin[5], Martin F. Sarott [6], Morgan Trassin [6], Maximilian Oezkent [7], Chen-Hsun Lu [7], Kevin-P. Gradwohl [7], Thomas Perrault [8], Lala Habibova[1], Giorgia Marcelli[1], Marcela Giraldo[6], Jan Vermant [2], Lukas Novotny [5], Martin Frimmer[5], Mads C. Weber [8] & Sebastian Heeg [1] ✉

Raman spectroscopy enables the non-destructive characterization of chemical composition, crystallinity, defects, or strain in countless materials. However, the Raman response of surfaces or thin films is often weak and obscured by dominant bulk signals. Here we overcome this limitation by placing a transferable porous gold membrane, (PAuM) on the surface of interest. Slot-shaped nanopores in the membrane act as plasmonic antennas and enhance the Raman response of the surface or thin film underneath. Simultaneously, the PAuM suppresses the penetration of the excitation laser into the bulk, efficiently blocking its Raman signal. Using graphene as a model surface, we show that this method increases the surface-to-bulk Raman signal ratio by three orders of magnitude. We find that 90% of the Raman enhancement occurs within the top 2.5 nm of the material, demonstrating truly surface-sensitive Raman scattering. To validate our approach, we quantify the strain in a 12.5 nm thin Silicon film and analyze the surface of a $LaNiO_3$ thin film. We observe a Raman mode splitting for the $LaNiO_3$ surface-layer, which is spectroscopic evidence that the surface structure differs from the bulk. These results validate that PAuM gives direct access to Raman signatures of thin films and surfaces.

Surfaces play a crucial role in a wide field of sciences and industrial applications. Surface effects dominate catalytic reactions to synthesize chemicals[1], the interaction of a biological cell with its environment[2], or the electric and thermal conductivity between elements in electronics[3]. Accordingly, there is a large number of experimental techniques for surface analysis with varying advantages and disadvantages, and better approaches to characterize surfaces are continuously being sought after[4]. Raman spectroscopy, the inelastic

[1]Institut für Physik und IRIS Adlershof, Humboldt-Universität zu Berlin, 12489 Berlin, Germany. [2]Soft Materials, Department of Materials, ETH Zürich 8093 Zürich, Switzerland. [3]Institute of Electromagnetic Fields (IEF), ETH Zürich 8092 Zürich, Switzerland. [4]Laboratory of Thermodynamics in Emerging Technologies Department of Mechanical and Process Engineering, ETH Zürich 8092 Zürich, Switzerland. [5]Photonics Lab, ETH Zürich 8093 Zürich, Switzerland. [6]Department of Materials, ETH Zürich 8093 Zürich, Switzerland. [7]Leibniz-Institut für Kristallzüchtung, 12489 Berlin, Germany. [8]Institut des Molécules et Matériaux du Mans, UMR 6283 CNRS, Le Mans Université, 72085 Le Mans, France. ✉e-mail: sebastian.heeg@physik.hu-berlin.de

scattering of light by vibrations or phonons, is ideal to study the structure of surfaces since their atomic registry differs from the bulk of the material and may additionally be modified by terminations. This leads to changes in the frequency of the Raman active vibrations or to peak-splitting as a result of a change in symmetry[5–7]. However, the study of surfaces and thin films by Raman spectroscopy is notoriously difficult as light typically penetrates several micrometers into the material. The material's overall Raman response is, therefore, dominated by the bulk. In contrast, the Raman signals of the surface are orders of magnitude weaker and mostly go undetected. Hence, obtaining Raman signals of thin films supported by or embedded in a bulk carrier requires a minimal film thickness of several tens to hundreds of nanometers. Raman signatures of surfaces are often not observed at all[8,9].

One approach to enhance the Raman signals stemming from material surfaces is tip-enhanced Raman spectroscopy (TERS), where a plasmonic hotspot at the apex of a metal tip scans over a surface[10–12]. The enhancement, however, occurs only at one small spot and bulk Raman signals are recorded together with the TERS signal. In addition, TERS remains a challenging technique, such that its use to probe surfaces or thin films is rarely reported. Probing surface and thin-film Raman signals has also been addressed by the mathematical decomposition of a large stack of spectra. To do so, multiple spectra of a sample are measured under varying conditions. An example can be altering the laser focal point with respect to the sample surface. Subsequently, a statistical analysis allows to decompose the spectra into contributions stemming from the substrate and from the surface, or the bulk and the thin film, respectively[13]. Another approach used to isolate the Raman signals of thin oxide films is to preferentially grow these films on those bulk substrates that minimally interfere with the thin film's Raman signals[9]. Overall, these techniques to obtain Raman signals of surfaces and thin films are limited in their sensitivity and mostly require dedicated equipment and complex data analysis.

An alternative route to obtain surface or thin-film Raman signals is reducing or suppressing the dominant bulk Raman signal. Ultraviolet (UV) Raman spectroscopy employs a UV laser taking advantage of the shallow penetration-depth of the UV-light into many materials[14,15]. However, the penetration of the laser into the material is still in the order of hundreds of nanometers, and can only be reduced further for materials with suitable band gaps[15]. Moreover, the damage threshold of many materials to UV light is low and the need for a UV laser and the appropriate optics limits the application of UV Raman[16].

In coherent anti-Stokes Raman scattering (CARS), a clever choice of layer thickness in samples consisting of stratified media enables the suppression of bulk Raman signals through destructive interference[17]. This approach, however, requires the Raman scattered light to be coherent through the use of pulsed lasers. Hence, it cannot be employed in standard Raman measurements with continuous-wave excitation.

Outside the Raman spectroscopy community, the idea of suppressing bulk signals to access surface signals is actively pursued[18]. A photonic waveguide, for example, can achieve a high surface-to-bulk molecular fluorescence ratio, which is useful in bioimaging, sensing, sequencing, and physical chemistry characterization[19]. The core idea is to use the highly localized and strongly distance-dependent evanescent fields of the waveguide to primarily excite and collect fluorescence from a surface, while the bulk signal is neither excited nor collected. This increases the surface-to-bulk fluorescence ratio by several orders of magnitude.

The ideal method to access surface or thin-film Raman signals should combine the enhancement of surface Raman signals with the suppression of bulk Raman signals. Plasmonic membranes, thin metallic films with nanoscale slots (i.e., nanopores), can deliver this combined functionality[20]. Upon resonant excitation, the slots in the membrane act as plasmonic slot antennas that harbour localized and enhanced near-fields, which rapidly decay outside the pore within a few nanometres. Placed on a material, the slot's near-fields interact primarily with the surface of the material and enhance its Raman signals. Away from the nanopore the metallic membrane reflects incident fields and bulk Raman signals, which effectively suppresses the bulk Raman signal. Using plasmonic membranes for surface Raman signal enhancement and bulk Raman signal suppression is very different from the typical use case of nanoporous gold structures including plasmonic membranes, which is the detection of trace amounts of molecules by surface-enhanced Raman scattering (SERS)[21–29]. Classical SERS requires hotspots that detect Raman signals of molecules ideally down to the single molecule level. Bulk suppression, on the other hand, is not required in SERS since there is no bulk molecule Raman signal that needs to be suppressed. This means that plasmonic membranes and nanoporous gold structures that work well for SERS may not be suitable for detecting surface Raman signals, which require nanopores that are located very close to the surface underneath. We have recently introduced transferable, easy-to-manufacture, and flat porous gold membranes (PAuM) with nanoscale pores acting as plasmonic slot antennas[30]. It was shown that individual pores feature local Raman enhancement factors up to $10^4$ to $10^5$ and sustain high excitation powers ($10^6$ W cm$^{-2}$), which makes them the ideal plasmonic structure to study surfaces and thin films with Raman spectroscopy.

Here, we use porous gold membranes to enhance the surface Raman signal and to simultaneously suppress the bulk signal of the sample. Using wavelength-dependent Raman spectroscopy, we show that PAuM enhances the surface-to-bulk Raman signal ratio by up to three orders of magnitude. Combining experiment and simulation, we reveal that the enhancement decays exponentially in the material such that 90% of the enhanced signal occurs within the top 2.5 nm. Hence, our approach enables highly surface-sensitive Raman spectroscopy for weak or bulk-obscured Raman signals. We directly apply this technique to study a strained Si top layer of a Si/SiGe heterostructure and the surface of a 20 nm LaNiO$_3$ thin film. Our technique reveals the characteristic phonon softening of the 12.5 nm strained Si film in agreement with X-ray reflectivity measurements[31]. Additionally, we find specific Raman signatures of the surface of the 20 nm LaNiO$_3$ thin film that differ from the Raman response of the entire film, in line with theoretical predictions and experimental observations using a scanning tunneling microscope[32]. Our work demonstrates therefore the potential of PAuM-supported Raman spectroscopy to study surfaces and thin films.

## Results

Our paper is structured as follows: First, we introduce the PAuM manufacturing and its working principle. Second, we demonstrate the surface enhancement and bulk suppression of Raman signals by a wavelength-dependent study with graphene as a model surface. Third, we unravel the depth dependence of the Raman response in experiments and simulation. To do so, we probe graphene sheets buried at various depths from the surface. Finally, we showcase the benefit of PAuM by quantifying strain in a 12.5 nm Si thin film and by revealing the Raman signature of a 20 nm LaNiO$_3$ film's surface.

### Manufacturing and working principle of PAuM

Figure 1a illustrates the key idea of this work: Without PAuM, a laser penetrates several multiples of its wavelength into the material, limited by absorption and focal depth. The Raman scattered signal, therefore, originates primarily from within the bulk. The surface Raman signal remains weak or non-detectable due to the vanishingly small scattering volume of the surface. In contrast, using PAuM, the surface Raman signal is drastically enhanced compared to the bulk Raman signal. This results from two simultaneous effects: 1) local plasmonic enhancement within the metallic nanopores and 2) the suppression of the laser

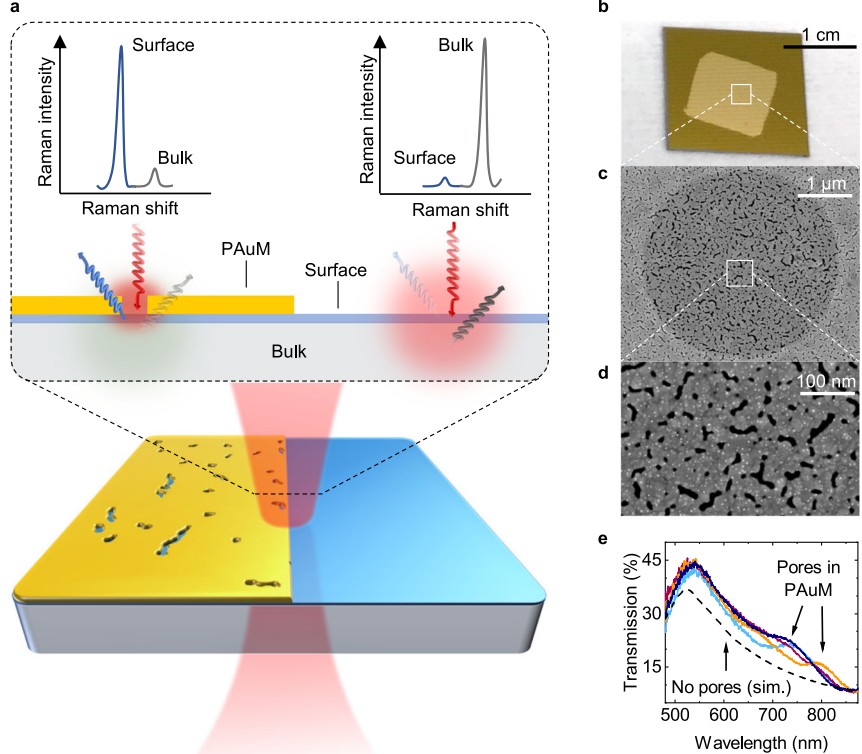

**Fig. 1 | Principle of bulk-suppressed and surface-sensitive Raman scattering by nanoporous Au membranes. a** Upon resonant excitation (red), nanopores in the gold membrane enhance the Raman signal of the surface (blue) while suppressing the bulk Raman signals (grey), which increases the surface-to-bulk Raman signal ratio by orders of magnitude. **b** Photographic image of a 20 nm PAuM transferred on a Si/Si$_3$N$_4$ with 4 $\mu$m holes (see Methods), forming freestanding membrane-like structures, as shown in the SEM image in **c, d**. The nanopores are visible as dark irregular, slot-like, and circular features in **d**. **e** Optical transmission of a freestanding PAuM measured. The dashed line corresponds to a simulated non-porous Au film with 20 nm thickness. The plasmonic resonances of the nanopores give rise to the increased transmission, measured at different locations (light blue, dark blue, purple, and orange lines) from 650 to 850 nm, compared to the simulation.

penetration into the bulk and of residual bulk Raman signal reaching the detection pathway.

Our non-continuous membranes are formed by evaporation of a 20 nm gold film on a 30 nm SiO$_2$/Si wafer. For details of the fabrication and membranes of other thicknesses, we refer to our previous work that introduces the membrane[30]. The membrane is subsequently transferred onto the sample of interest[30] (see Methods and Supplementary Note 1). To characterize the PAuM used in this work, we transfer a 1 cm$^2$-sized membrane on a Si/Si$_3$N$_4$ chip bearing arrays of 4 $\mu$m holes (Methods) as shown in the optical microscope image in Fig. 1b. The scanning electron microscope (SEM) images in Fig. 1c, d reveals that the PAuM spans over the circular hole as a freestanding, mechanically stable membrane[30]. The pores in the PAuM are visible in the SEM images as dark spots and come in various shapes and sizes. The majority of pores is round or slot-like and below 100 nm in size (see Supplementary Note 1). We have demonstrated that the nanopores in the PAuM act as plasmonic nanoslot antennas[30]. The nanopores harbour intense light fields upon excitation at their plasmonic resonance. The energy of the plasmonic resonance depends on the shape and aspect ratio of the individual pores. The highest field enhancement occurs for narrow slot-like pores with an excitation polarized perpendicular to the pores' long axis, see ref. 30.

In Fig. 1e, we compare the optical transmission of freestanding 20 nm PAuM to the transmission of a simulated gold film without nanopores of the same thickness. The general shape of the optical transmission is in good agreement with the simulated results (dashed line). However, deviations occur from 650 to 850 nm with an increased transmission probability. Such increased transmission is expected when the nanopores of PAuM are resonantly excited and act as nanoantennas[20]. The colored lines correspond to individual transmission measurements at different sample positions, and their varying deviations from the simulated trend indicate the varying geometries of the nanopores. We conclude that Raman enhancement can be expected in the spectral range from 650 to 850 nm in agreement with our previous study[30]. The random nature of pore geometries in both - dimension and orientation - provides a substantial number of pores that will resonantly couple to an incident laser light with arbitrary polarization and wavelength.

## Measurement of surface enhancement using graphene probes

Graphene is ideal as a model material to test and quantify surface-sensitive Raman scattering as it can mimic a surface or thin film due to its two-dimensional nature[33–35]. Furthermore, the Raman spectrum of graphene is well understood and the main Raman features are intense and independent of excitation wavelength as well as polarization[36]. Any dependence of the Raman features of graphene interfaced with our porous Au membrane can hence be attributed entirely to nanopore interaction.

To probe the interaction and enhancement of PAuM with a surface, we place a graphene sheet, acting as a test surface, on a flat 300 nm SiO$_2$/Si substrate, see Supplementary Note 2. A PAuM is transferred on top of this model system (see Methods) to cover the graphene sheet partially as sketched in Fig. 2a. In this way, we can probe the two effects that contribute to surface-sensitive Raman scattering: First, the surface enhancement by the plasmonic nanopores of the PAuM via the graphene Raman signal and second, the bulk-signal suppression by monitoring the Raman signal of the Si substrate with

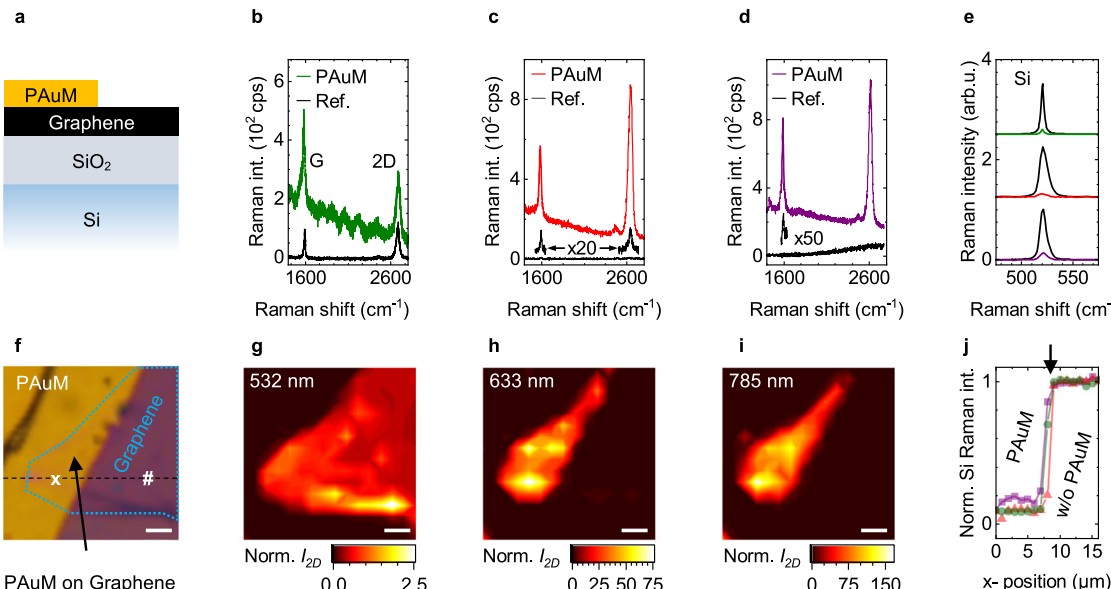

**Fig. 2 | Graphene as model surface to probe surface-sensitive Raman scattering by nanoporous Au membranes. a** Sample schematic where a graphene flake on a 300 nm SiO$_2$ on Si substrate is partially covered by a PAuM. Raman spectra of the bare graphene (black, reference) compared to spectra from graphene plus PAuM for (**b**, green) 532 nm, (**c**, red) 660 nm, and 785 nm (**d**, purple) excitation. The background in **b–d** stems from gold luminescence with a modulation due to etaloning for 532 nm. **e** First-order Silicon Raman peak with PAuM for 532 nm (green), 660 nm (red), and 785 nm (purple) normalized and compared to the corresponding reference Raman spectra (black) without PAuM. The spectra for each wavelength are offset for clarity. **f** Microscope image of graphene flake (dashed turquoise)

partially covered by PAuM (yellow). × (with PAuM) and # (reference) mark the locations of the spectra shown (**b–e**). The dashed line marks the location of the spatial profile shown in **j**. Spatial Raman maps of the graphene 2D mode for **g** 532 nm, **h** 660 nm, and **i** 785 nm excitation. For each excitation wavelength, the intensity is normalized to the 2D intensity of bare graphene reference with interference effect taken into account. **j** Spatial profile of the normalized silicon Raman intensity along a horizontal line (dashed in **f**) through the locations x and # as indicated in **f** for all three excitation wavelengths. The arrow marks the edge of the PAuM. The scale bar in **f–i** is 2 μm.

and without the membrane. Figure 2f shows a light microscope image of the PAuM-graphene stack on a substrate. The PAuM appears as the yellow region, and the monolayer graphene flake is marked by the dotted line. As can be seen in the image, the graphene flake is partially covered by the PAuM.

First, we compare individual Raman spectra of PAuM-covered and uncovered graphene for three excitation wavelengths in Fig. 2b–d. The uncovered graphene serves as a reference and the corresponding reference spectra are shown in black for each excitation wavelength, where × and # in Fig. 2f mark the measurement positions. For 532 nm excitation, Fig. 2b, the G and 2D Raman modes of graphene show comparable intensities for PAuM-covered (green) and uncovered graphene (black). This behavior is expected since the nanopores in the membrane are not in resonance with the 532 nm laser excitation[30]. The 2D-to-G intensity ratio further confirms that the graphene in these measurements is indeed a monolayer[37,38].

For 660 nm excitation, the Raman spectrum for PAuM-covered graphene (red) is substantially enhanced when compared with uncovered graphene (black) in Fig. 2c. The enhancement is even more pronounced for 785 nm excitation, Fig. 2d. In this wavelength range, plasmonic enhancement from the pores comes into play, in good agreement with our transmission data, Fig. 1e, and previous works[30]. For a quantitative analysis of the enhancement, we take care of potential reflection effects at the graphene-Si/SiO$_2$ interface[39]. After excluding interference effects (Supplementary Notes 2 and 3), we find enhancement factors between 33 and 165 for the G and 2D modes (Table 1). Note that the 2D-mode intensity for bare graphene and 785 nm is below the noise level. We therefore use the noise level to approximate the 2D-mode intensity.

The spatial Raman maps shown in Fig. 2g–i trace the intensity of the 2D Raman mode of graphene $I_{2D}$, normalized to the uncovered graphene reference, for 532, 660, and 785 nm excitation, respectively.

Interference effects are accounted for in all maps. The Raman map in Fig. 2g confirms the negligible effect of the PAuM on the Raman response for 532 nm excitation. In contrast, the Raman maps for 660 and 785 nm excitation show the striking enhancement by the PAuM. Clearly, the enhancement only occurs in areas of PAuM-covered graphene. Local variations in the enhancements reflect the random distribution of the plasmonic nanopores in the membrane with respect to geometry and orientation.

Second, we demonstrate the suppression of the bulk/substrate Raman signal by the PAuM. To do so, we make use of the silicon substrate 300 nm below the PAuM, where the silicon Raman signals are not enhanced. We compare the first-order Raman peak of silicon at 521 cm$^{-1}$ with (colored) and without a PAuM (black) for our three excitation wavelengths of 532 nm (top), 660 nm (middle), and 785 nm (bottom) in Fig. 2e. The spectra indicate a clear suppression of the silicon Raman signal with PAuM for all wavelengths. This agrees with the attenuation expected for the incoming laser light and the Raman scattered light for a non-porous membrane of comparable thickness, see Fig. 2c. A line scan across the edge of the PAuM (Fig. 2j) reveals a constant silicon Raman signal for all wavelengths on either side of the edge. The experimentally observed suppression by the PAuM amounts to a factor of 10 for 532 and 660 nm, and 6.6 for 785 nm, see Table 1. Since the Si Raman signal without PAuM used as reference is affected by interference, the bulk Raman signal suppression is a factor 5 to 8 higher than the experimentally obtained values, see Supplementary Note 4. This brings the suppression closer to the values expected from our transmission experiments, see Fig. 1e. The exact magnitude of bulk Raman signal suppression depends on the exact geometry of the sample investigated. We therefore consider it most adequate to provide the experimental values as a lower bound for bulk Raman signal suppression by our PAuM. The product of surface Raman enhancement and bulk-suppression, which is the primary figure of merit for

**Table 1 | Enhancement factors, bulk suppression, and surface-enhancement multiplied by bulk-suppression as figure of merit for the performance of PAuM for 660 and 785 nm excitations**

| Wavelength | Enh. G | Enh. 2D | Bulk suppression | Surface-enhancement × bulk-suppression |
|---|---|---|---|---|
| 660 nm | 72 | 69 | 10 | 690 to 720 |
| 785 nm | 33 | 165 | 6.6 | 220 to 1100 |

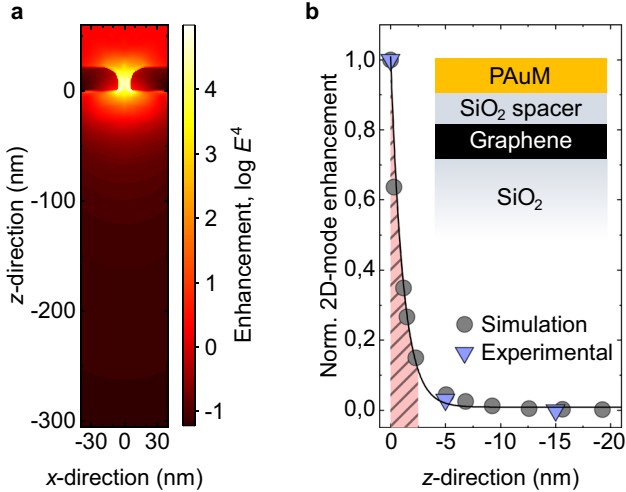

**Fig. 3 | Enhancement of Raman signal as a function of distance from the surface. a** Simulated $E^4$ enhancement (log scale) in the $xz$-plane for a 22 nm thick gold membrane with a 10 nm by 68 nm slot on $SiO_2$. **b** The experimental (blue triangle) and simulated (grey circles) enhancement for the graphene 2D Raman mode are shown as a function of $SiO_2$ spacer thickness between the porous Au membrane and graphene. The black line is an exponential fit to the simulated values. The light red dashed area marks the volume within which 90% of the total Raman enhancement occurs. The simulated enhancement is normalized to the value at $z = -0.35$ nm, which is equivalent to the thickness of single layer graphene.

surface-sensitive Raman scattering as suggested here, then amounts to values between 220 and 1100, see Table 1.

## Depth dependence of Raman enhancement by PAuM

Next, we investigate the effective Raman enhancement of the material below the PAuM as a function of depth by simulation and experiment. Since nanopores act as slot antennas[30], we simulate the Raman enhancement by a prototypical plasmonic nanoslot (10 nm × 68 nm) similar in shape and size to nanopores found in the PAuM using a finite element solver JCMsuite (version 5.2.0) for Maxwell's equations (see Supplementary Fig. 5 for the corresponding extinction and scattering cross sections). We assume a wavelength of 660 nm for excitation and 800 nm for the Raman scattered light of the graphene 2D-mode (see Methods and Supplementary Note 5). Figure 3a depicts the simulated enhancement in the $xz$-plane along the nanoslot's short axis ($x$-direction). We then extract the average enhancement in the entire $xy$-plane from our simulation as a function of depth $z$. To obtain the average, we do not only consider the areas directly under the nanoslots but also the area around it. This includes an area 15 times larger than the area of the slot and accounts for the fact that each nanopore is surrounded by a continuous gold membrane segment, see Fig. 1b. We plot the average Raman enhancement of the graphene 2D-mode versus $z$ in Fig. 3b and find that the enhancement drops sharply with an increasing distance from the nanoslot. The decay is described by an exponential function $e^{-z/\tau}$ with $\tau = 1.1$ nm. This means that 90% of the total Raman enhancement occurs within the first 2.5 nm below the nanoslot.

In the next step, we probe the field enhancement as function of distance from our PAuM experimentally. To do so, we sputter 5 and 15 nm $SiO_2$ spacers on graphene and subsequently transfer PAuM on top as sketched in Fig. 3b, see Supplementary Notes 6. We plot the enhancement of the graphene 2D-mode with and without the two different spacers together with the simulation data in Fig. 3b. We find that the experimental enhancement is in excellent agreement with the simulation. This finding confirms that the PAuM-enhanced Raman spectroscopy allows for truly surface-sensitive Raman scattering, with an effective enhancement depth of less than 5 nm.

The good agreement between our simulation and the experimental data for graphene allows us to infer more general properties of the surface enhancement provided by our PAuM. Extended simulations indicate an equally fast decay with distance from the PAuM for the entire relevant Raman frequency range of a few to 2500 cm⁻¹ (Supplementary Note 4). Furthermore, we find the strongest enhancement and the sharpest enhancement decay for nanopores with plasmonic resonances having high quality-factors. High quality-factors are inherent to narrow pores, i.e., gap features < 10 nm[30]. Wider pores with lower aspect ratios feature lower quality-factors, which leads to a reduced enhancement and greater decay constants $\tau \approx 5$ nm, equivalent to 90% of the total Raman enhancement within the first 11.5 nm. The approach to achieve the best surface-to-bulk enhancement ratio of the Raman signal is therefore to identify the highest Raman enhancement within a spatial map (see for example Fig. 2) and evaluate its spectrum.

## Probing strain in a Si/SiGe heterostructure

To demonstrate the benefit of surface-sensitive Raman scattering by PAuM, we quantify strain in a 12.5 nm thin Si quantum well layer coherently grown on relaxed $Si_{0.7}Ge_{0.3}$. These SiGe heterostructures are utilized as electron spin qubits in gate-defined silicon quantum dots and represent a promising and fast-developing technology for quantum computing[40,41]. This material platform allows scalable fabrication of nuclear spin-free heterostructures by utilizing isotopically enriched Si and Ge (such as ²⁸Si and ⁷⁶Ge) due to its compatibility with standard silicon processing.

However, one challenge is the lifting of the conduction band valley degeneracy, which is partially achieved by biaxial strain in the Si quantum well layer. In particular, the strain determines the uniformity of the energy landscape as experienced by large arrays of qubits, which is a challenge for shared gate control architectures. It is therefore of high importance to quantify the strain in the Si film and understand the sources of strain inhomogeneity in detail. Since the quantum well layers are typically only 10 nm, the use of Raman spectroscopy to extract strain is strongly limited. Previous studies employed high-resolution X-ray diffraction techniques[42] or UV Raman spectroscopy, since using UV light reduces the penetration depth, allowing to collect Raman signals from the sample surface[43,44]. Surface-sensitive Raman scattering by PAuM introduced in this work constitutes a promising tool to probe strain in Si quantum well layers, which does not require UV Raman instrumentation.

A test structure with a 12.5 nm thick strained ²⁸Si layer was grown on a ²⁸$Si_{0.7}$⁷⁶$Ge_{0.3}$ barrier by molecular beam epitaxy (MBE) on a relaxed $Si_{0.7}Ge_{0.3}$ virtual substrate on Si (Methods). A PAuM is then transferred to the sample, partially covering the strained ²⁸Si thin film. An illustration of the entire structure is shown in Fig. 4a.

Figure 4b,c depicts the Raman spectra of strained ²⁸Si with (colored) and without (black) PAuM for 638 nm and 785 nm excitation, respectively. The dominant Raman peak at 503 cm⁻¹ arises from the Si-Si local mode of $Si_{1-x}Ge_x$[45]. Its frequency lies in between the one of the $Si_{0.77}Ge_{0.23}$ layer below the strained ²⁸Si thin film (see Fig. 4a) with a Si-Si Raman peak at 505.3 cm⁻¹ and the relaxed $Si_{0.7}Ge_{0.3}$ with a Si-Si Raman peak at 500.6 cm⁻¹[46]. The Si-Si peak at 520.7 cm⁻¹ from bulk Si is far more intense in the Raman spectra at 785 nm due to the higher

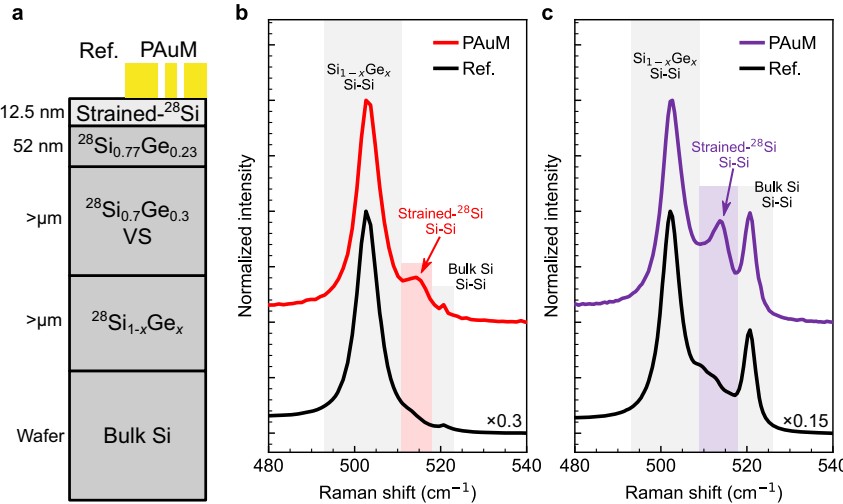

**Fig. 4 | Quantifying strain in a strained $^{28}$Si thin film by surface-senstive Raman scattering. a** Sketch of the strained $^{28}$Si thin film and its substrate (see main text for details, VS labels the SiGe virtual substrate). The compound is partially covered by a PAuM. Raman spectra of the strained $^{28}$Si thin film with PAuM (red line in **b** and purple line in **c**) and without PAuM (black lines in **b** and **c**), collected using **b** a 638 nm and **c** a 785 nm laser. Important Raman bands have been highlighted for visual guidance.

penetration into the sample. The Si-Si peaks from $Si_{1-x}Ge_x$ and bulk Si are suppressed in the presence of a PAuM on top (colored lines in Fig. 4b,c). We infer a bulk suppression factor of about 3.5 for 638 nm and 6.5 for 785 nm excitation. Note that these values are below those expected from our transmission data in Fig. 1e, which we attribute to interference at the interfaces between the layers of the heterostructure as previously discussed.

Raman spectra from regions covered by PAuM feature an intense peak at 514 cm$^{-1}$ that agrees to within 0.5 cm$^{-1}$ for 638 and 785 nm excitation. This Raman feature is absent when measuring without the PAuM and unequivocally corresponds to the Si-Si peak of strained Si[31]. The appearance of this specific Raman peak originating from the 12.5 nm strained Si thin film, demonstrates surface-sensitive Raman spectroscopy enabled by our PAuM. Moreover, the PAuM enhancement is clearly higher when measuring for 785 nm excitation, in agreement with our measurements on graphene. The phonon softening of the Si thin film Raman peak compared to bulk Si is a clear sign of tensile strain[47]. We calculate the strain using the Raman strain shift coefficient obtained by Wong et al.[31] (see Methods section for further details), and obtain an average tensile strain of 0.95% ± 0.02%, in good agreement with the value of 1.01% found using XRR, see Methods and Supplementary Fig. 12.

### Probing structural properties of a thin-film oxide surface

As a second application, we use our porous gold membranes to investigate a crystalline surface. We choose a complex oxide, a LaNiO$_3$ thin film on LaAlO$_3$, as showcase material. Complex oxides are particularly suitable because their physical properties are highly sensitive to structural distortions[48–50]. LaNiO$_3$ illustrates this well, as it is one of the few conducting perovskite-type materials and therefore an important electrode material for perovskite-type heterostructures[51]. However, its conductivity is thickness-dependent, where below a thickness of 3 unit cells, LaNiO$_3$ even exhibits insulating behavior. These conductivity changes are attributed to structural inhomogeneities in the thin films. More specifically, the bulk and the surface of a LaNiO$_3$ film show different degrees of structural distortions[32]. A previous Raman study of LaNiO$_3$ thin films confirmed the structural changes with film thickness[13]. However, these Raman spectroscopic measurements would only give information about the average structures and did not distinguish between surface, bulk or heterointerface.

Here, using PAuM-enhanced Raman spectroscopy, we specifically target the surface structure of LaNiO$_3$ thin films to directly observe structural distortions.

A 20 nm-LaNiO$_3$ thin film has been epitaxially grown on a (100)$_{pc}$-oriented LaAlO$_3$ substrate by pulsed laser deposition, see Methods and Supplementary Fig. 13. Subsequently, a PAuM is transferred onto the thin film sample. As bulk, LaNiO$_3$ and LaAlO$_3$ crystallize in a rhombohedral perovskite-type structure with the space group characterized by anti-phase rotations of the octahedra, $a^-a^-a^-$ in Glazer's notation (see Fig. 5a). The structure gives rise to five Raman-active vibrational modes $\Gamma_{Raman} = A_{1g} + 4\,E_g$[52]. Epitaxially strained LaNiO$_3$ on LaAlO$_3$ stabilizes a monoclinic structure with the space group $C2/c$ ref. 13. However, for simplicity reasons and the similarity of the Raman spectra of epitaxially strained LaNiO$_3$ films and bulk LaNiO$_3$, we retain the notations of the rhombohedral bulk symmetry.

For our Raman spectroscopy experiment, we use an excitation wavelength of 785 nm, as it shows the best performance with PAuM, see Table 1 and ref. 30 (see Supplementary Note 7 for the spectra under 660 nm excitation). Figure 5b shows the Raman spectra of LaNiO$_3$ on LaAlO$_3$ with (purple) and without PAuM (black). The light-blue and gray boxes correspond to regions with vibrational bands of LaAlO$_3$ and LaNiO$_3$, respectively. The Raman spectrum measured without PAuM is in perfect agreement with literature data[13,53]. Yet, the comparison of the spectra with and without PAuM reveals a number of striking differences: With the PAuM, the signal of the LaAlO$_3$ substrate is barely visible and can only be approximated as shoulder-like features in the region between 100 and 160 cm$^{-1}$. Furthermore, the LaNiO$_3$-$E_g$ mode around 400 cm$^{-1}$ exhibits a prominent difference. In the spectrum without PAuM, we find a single, albeit asymmetric, peak at 412 cm$^{-1}$. In contrast, the spectrum with PAuM has two distinct features centered at 389 and 413 cm$^{-1}$, as highlighted by the red box. For a high-resolution Raman spectrum of this region see Supplementary Note 7. The $A_{1g}$ mode of LaNiO$_3$, on the other hand, shows only a minor shift of ≈3 cm$^{-1}$ from 213 cm$^{-1}$ without PAuM to 216 cm$^{-1}$ with PAuM. Note, at low frequencies, the spectrum with PAuM is characterized by an intensity increase. We assign this increase to the onset of intense low frequency $E_g$-mode of LaNiO$_3$ at 74 cm$^{-1}$ below the spectral cut-off of the measurement setup.

Our results allow for a number of interesting conclusions: First, in the presence of the PAuM, the thin film signal is strongly enhanced with respect to the LaAlO$_3$ substrate signal. Second, the $E_g$-mode

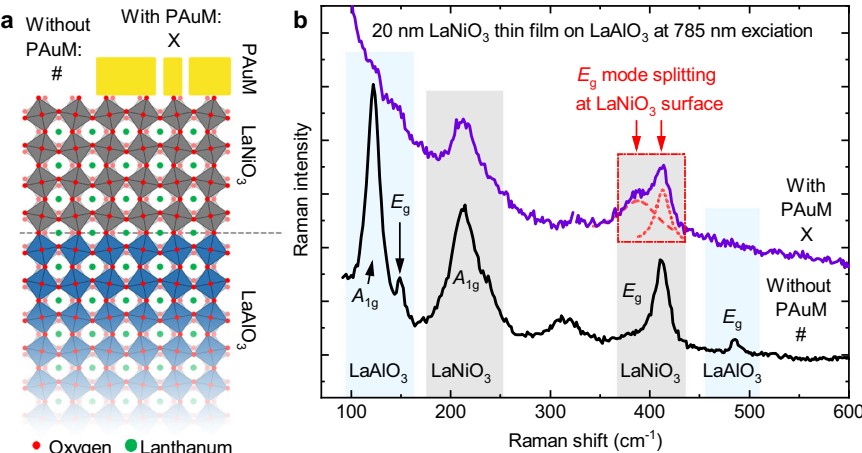

**Fig. 5 | Surface-sensitive Raman scattering of a LaNiO₃ thin film's surface.**
**a** Sketch of the perovskite-type LaNiO₃ thin film on a $(100)_{pc}$-oriented LaAlO₃ substrate. The compound is partially covered by a PAuM. The symbols X and # indicate the measurement positions of the spectra in **b** with and without PAuM, respectively. **b** The Raman spectrum of a LaNiO₃ thin film on LaAlO₃ with PAuM (purple line) and without (black line) PAuM are depicted. Important Raman bands are highlighted for LaNiO₃ (light blue) and LaNiO₃ (grey). With PAuM, the formation of a shoulder peak at 389 cm⁻¹ (see red box and red arrows) next to the main peak at 413 cm⁻¹ indicates the distinct structural difference of the surface layer compared to the bulk of the LaNiO₃ thin film.

splitting in the Raman spectrum demonstrates that the PAuM-enhanced Raman signal does not represent an average of the vibrational signal of the entire film. Rather, the PAuM-enhanced Raman signal stems primarily from the surface layer, namely the first few nanometers where the PAuM enhancement is most effective. This explains the seemingly lower signal-to-noise ratio for the spectral response with PAuM compared to the spectrum without PAuM. Using PAuM, the overall scattering volume has been reduced dramatically. The signal stems only from the positions of the pores, where the surface signal is strongly enhanced. Importantly, this Raman signal is independent of the film thickness (see Raman spectra of 7 and 14 nm films in Supplementary Note 7), which unambiguously confirms that it arises from the film's surface.

From the material perspective, we know that the crystal structure changes significantly within these first few nanometers[32]. A look at the vibrational patterns reveals further details of those surface distortions. The $A_{1g}$ and $E_g$ around 213 and 403 cm⁻¹ of bulk-like LaNiO₃ correspond to an octahedral tilt and bending vibration patterns, respectively[52,53]. Therefore, we suggest that deformations of the octahedra, permitted by the $C/2c$ symmetry, dominate the structural changes in the surface region over changes of the octahedron tilt-angles.

Note that the PAuM plasmonic enhancement of the Raman spectra of oxide materials differs from graphene, our model surface previously, where the primary scattering volume is reduced. Hence, the Raman spectra with a PAuM have a lower total intensity and signal-to-noise ratio than without the PAuM, despite the enhancement of surface Raman signals (Fig. 5). Overall, our findings demonstrate that enhancement of the Raman signal by PAuM allows the effective extraction of the Raman response of an oxide surface. This promising spectroscopy-based access to the structure reveals major structural changes at the film surface compared to the bulk of the film, in agreement with the monoclinic symmetry of the film.

## Discussion

The improvement of the surface-to-bulk Raman intensity ratio by up to three orders of magnitude is the primary quality of the PAuM. Moreover, PAuM can sustain excitation power densities up to 10⁶ W cm⁻² for 785 nm excitation without structural damage. This exceeds the threshold of classical plasmonic structures by two orders of magnitude[30] and further increases the detectable Raman signal from a surface. Furthermore, surface-sensitive Raman measurements with

PAuM equally function at cryogenic temperatures (see Supplementary Note 8). Hence, PAuM-enhanced Raman spectroscopy may give access to temperature-driven phase transitions of a surface layer[54].

The geometrical randomness of the pores in our PAuM always provides pores with suitable enhancement for an arbitrary combination of excitation wavelength, polarization, and refractive index of the probed material, see Supplementary Note 4. We can further tune the ratio between surface enhancement and bulk suppression by altering the PAuM thickness. Thicker PAuM lead to fewer nanopores and reduced transmittance, thinner PAuM to more nanopores and increased transmittance. Thermal annealing at moderate temperatures provides additional means to alter the shape of the nanopores in our PAuM.

The price to pay for surface sensitivity using PAuM is that the polarization and wavelength dependence of the Raman measurements are dominated by the plasmonic resonance of a nanopore. Light fields inside a nanopore are polarized primarily along the nanopore's short axis and measured Raman intensities predominantly follow the nanopores' plasmonic resonances. This means that the use of polarization and wavelength-dependent Raman measurements to study the properties of a surface or thin film is rather limited. However, this limitation is not specific to PAuM since it occurs in any structure that supports plasmon-enhanced spectroscopy. In addition, metallic nanostructures for plasmonic enhancement may also entail the photoinjection of hot electrons into the material. This effect is particularly pronounced for small nanoparticles, rough interfaces, and nanogapped structures with extreme field confinement[55–57]. In contrast, our plasmonic membrane is an extended structure that also features a very smooth interface since it was fabricated on flat SiO₂, similar to plasmonic structures obtained by template stripping[58]. We conclude that our membranes will hardly undermine charge carrier momentum conservation, which is needed for efficient electron or hole transfer[56,57]. Nevertheless, the role of hot carrier injection in surface-sensitive Raman scattering by plasmonic membranes and its effect on the Raman response of the surface or film under study requires further clarification and will be subject to future work. A potential route to prevent hot carrier injection is inserting an insulating barrier, such as a monolayer of hexagonal boron nitride, between the PAuM and the surface[59].

PAuM can be transferred to non-flat, porous or flexible substrates, as for example done in previous work[30]. However, the surface enhancement effect is then limited to the contact points of the

membrane with the substrate, thereby reducing the overall signal strength (see Supplementary Note 9 for detailed discussion).

In general, surface-sensitive and bulk-suppressed Raman spectroscopy is not limited to PAuM. The defining features of a plasmonic structure to enable surface-sensitive Raman scattering are (i) the formation of a hotspot very close to the surface of interest, (ii) the suppression of bulk signals, and (iii) the ability to sustain laser irradiation without structural damage. While other structures such as nanoparticle films or isolated bowtie structures offer one or two of these defining features, none offer the combination of all three. In contrast, PAuM and other types of plasmonic membranes fulfill these requirements. Membranes with nanopores of defined geometry, for instance, can be tailored to independently maximize bulk suppression and surface enhancement for specific excitation wavelengths. However, manufacturing this type of structure requires dedicated instrumentation (e.g., focused ion beam) often employing time-consuming serial processes, whereby one pore is created at the time, limiting the size of the membrane to a few hundreds of $\mu m^2$. Our PAuM, on the other hand, are manufactured by simple evaporation on wafer-scale and provide broadband surface enhancement without cumbersome and laborious processing steps.

In conclusion, we have demonstrated that transferable nanoporous gold membranes enable surface-sensitive Raman spectroscopy. Slot-shaped nanopores in the membrane act as plasmonic hot-spots for enhanced Raman scattering. Simultaneously, the membrane nature of the PAuM suppresses bulk Raman signals. Using graphene as a model surface, we have shown an increase in the Raman surface-to-bulk ratio of a factor 1100. Simulations combined with Raman measurements on buried graphene samples showed that the enhancement drops exponentially with distance from the PAuM. 90% of the enhancement occurs within the top 2.5 nm of the probed material. To demonstrate the utility of our approach, we extracted the strain in 12.5 nm thin Si layer and applied it to an open scientific question – the spectroscopic analysis of the surface structure of LaNiO$_3$. By PAuM-enabled surface-sensitive Raman scattering of a LaNiO$_3$ thin film on LaAlO$_3$, we found major structural changes at the surface of LaNiO$_3$, which had not been observed by Raman spectroscopy to date. Surface-sensitive and bulk-suppressed Raman scattering, as introduced in this work, therefore extends the use of Raman spectroscopy as an analytical technique to study surfaces and thin films. Our approach is not limited to crystalline surfaces, but may also be employed to monitor surface-bound chemical reactions or to characterize biological membranes.

## Methods

### PAuM manufacturing and transfer

A non-continuous gold film of 20 nm is evaporated on a silicon/silicon dioxide (Si/SiO$_2$) Wafer with an oxide thickness of 30 nm at a rate of 0.2 nm s$^{-1}$ in an e-beam evaporator (Evatec AG)[30]. The wafer is cleaned using acetone, isopropyl alcohol, and oxygen plasma (600 W, 2 min, PVA TePla AG) to remove any contamination before the gold evaporation. The PAuM is subsequently coated with poly(methyl methacrylate) (PMMA) in Anisol (2 wt.%), followed by a floating etch in buffered hydrofluoric acid (BHF), releasing the PAuM/PMMA from the substrate. The PAuM/PMMA is subsequently rinsed by floating on deionised water for a total of 60 min, after which it is scooped with the target substrate and let dry. Oxygen plasma (for non-organic samples, 400 W for 4 min) or acetone/isopropyl alcohol baths (for organic samples) are used to remove the PMMA (see Supplementary Note 1 for details, including procedures to remove PAuM after Raman measurements). Freestanding PAuM for pore size characterization and transmission measurement are obtained using a pre-patterned Si/Si$_3$N$_4$ chip with arrays of 64 holes of 4 $\mu m$ size, as described by Celebi et al.[60].

### Optical transmission of PAuM

Transmission measurements were performed using a self-built setup by focussing white light from a broadband supercontinuum laser (NKT) with an objective (NA 0.9) on the freely suspended PAuM from the top. A long-distance objective (NA 0.7) placed below the PAuM collected the transmitted light, which was recorded with a Princeton Instruments Acton spectrometer. Reference spectra taken without the PAuM were used to subtract the background and eliminate any wavelength dependence of the spectrometer.

### Graphene synthesis and transfer

Graphene was synthesized on copper by chemical vapor deposition following Choi et al.[61]. Copper foil (Alfa Aesar N° 46986) was pretreated by ion beam milling (Argon 10 cm$^3$ STP min$^{-1}$, 10 min, 250 mA) or alternatively in ammonium persulfate (APS, 0.5 M for 5 min) to remove any surface contamination. Graphene was subsequently grown in a cold-wall chemical vapor deposition system (Aixtron Black Magic 4 inch) at 950 °C flowing 1500 cm$^3$ STP min$^{-1}$ Argon and 100 cm$^3$ STP min$^{-1}$ hydrogen at a pressure of 2 mbar for 30 min to anneal the surface. Flowing ethylene 25 cm$^3$ STP min$^{-1}$ for 2 min and 50 cm$^3$ STP min$^{-1}$ for 1 min led to the formation of a complete monolayer graphene. The sample was cooled to room temperature flowing 100 cm$^3$ STP min$^{-1}$ Argon. The graphene on copper is covered with PMMA (in Anisol, 2 wt.% by spin-coating, followed by a floating release etch in 0.5 M ammonium persulfate. The samples are rinsed in DI-water and transferred to the final substrate. PMMA is removed by acetone and isopropyl alcohol.

### Buried graphene sample preparation

Buried graphene samples for depth-dependent PAuM enhancement measurements are manufactured by transferring as-grown graphene to a substrate of Si/SiO$_2$ with an oxide thickness of 30 nm. Subsequently, SiO$_2$ is sputtered with a thickness of 5 and 15 nm. The sputtering conditions (Von Ardenne CS3206 Cluster) are 600 W power, 5 cm$^3$ STP min$^{-1}$ argon at 0.2 Pa total pressure.

### Numerical finite-element simulations

The gold membranes feature randomly oriented, shaped, and sized pores. It is not insightful to theoretically study many realizations of potential or actual pore shapes as they will vary randomly from membrane to membrane. Instead, we studied relatively simple and uniform pores in order to estimate the general performance of the membranes. Specifically, we studied round and elongated rectangular pores. This is guided by the idea that mainly dipolar modes will be excited within pores when excited from the far-field with a weakly focused beam. We used the finite element solver JCMsuite (version 5.2.0) for Maxwell's equations to study the nanopores theoretically. We computed scattering spectra to estimate resonance frequencies as well as nearfields to derive a prediction for the local Raman enhancement. In all simulations, plane waves were used as light sources, impinging from above (air side) orthogonally to the interface of the substrate in correspondence to the experiments. As general trends, we found that round pores provide significantly lower field enhancements as compared to elongated slots. Round pores also feature mainly resonance wavelength around 530 nm. Both effects can be understood by referring to Babinet's principle, i.e., spheres behave similarly to round holes and rods behave similarly to slots in a membrane: Rods will have variable resonance frequencies depending on their aspect ratio and higher quality factors, which gain higher local field enhancements. We concluded that most of a Raman signal stems from elongated slots and focused our analysis thereon, which is in excellent agreement with a previous study on the plasmonic properties of nanopores in our PAuM[30]. Further, we found that smaller pores yield higher Raman enhancements with a steeper decay into the substrate, i.e., stronger

bulk signal suppression. This is plausible: the smaller a particle (or pore) the smaller its scattering cross section and the higher its quality factor will be until the quasi-static limit is reached[62]. For elongated slots, we performed 3D simulations with a stacked layout, i.e., many layers of different permittivities. We made use of two mirror-symmetries, i.e., the computational effort reduces by approx. a factor of four. To avoid unphysical hotspots at sharp corners, we rounded all corners, also in the direction of stacking. Two orthogonally polarized plane waves (polarized along the short and long axis of the slot) have been used as light sources in these computations. Thus, all intensity enhancements shown here are mean values of the two polarizations. For the simulations, we used permittivities described in refs. [63–65]. Even, if we restrict ourselves to rectangular slots, the potentially interesting parameter room is still vast: factors like the substrate's permittivity, the thickness of the gold film, mode volume, and quality-factor could affect the predicted Raman enhancement. Therefore, we focus on a few examples, i.e., we use a gold-film thickness of 22 nm, and if not stated differently, we show results of a 68 nm × 10 nm slot, resonant at 720 nm (Supplementary Fig. 5).

## Raman spectroscopy
Raman measurements were performed on a Horiba LabRam Raman spectrometer and a Horiba Xplora spectrometer, both equipped with motorized stages. Laser powers were kept below 3 mW (100 × objective, NA 0.9) with integration times up to between 1 s and 30 min. Further Raman measurements were performed using a WiTec Apyron 300R Raman spectrometer (100 × objective, NA 0.9).

## Si/SiGe heterostructure growth
The $^{28}$Si/$^{28}$Si$_{0.7}$$^{76}$Ge$_{0.3}$ heterostructure was grown by molecular beam epitaxy on relaxed Si$_{0.7}$Ge$_{0.3}$ virtual substrates. The substrate was treated with a standard Si wet-chemical cleaning procedure before growth; involving degreasing for 5 min in acetone and isopropanol, 10 min in hot Piranha solution, 5 min in 0.5% HF solution, and a dip in deionized water. Furthermore, the sample was annealed in an ultra-high vacuum environment at 650 °C for 10 min. Isotopically pure $^{28}$Si and $^{76}$Ge source materials were applied (nuclear spin concentration < 100 ppm). The heterostructure was grown under controlled conditions utilizing a cryo pump and a liquid nitrogen cryo shroud, to stabilize growth pressure around $10^{-10}$ mbar. The nominally 50 nm of $^{28}$Si$_{0.7}$$^{76}$Ge$_{0.3}$ were grown at 400 °C while the 12.5 nm of strained $^{28}$Si layer were grown at 300 °C, with growth rates of 0.020 and 0.006 nm s$^{-1}$ for $^{28}$Si and $^{76}$Ge, respectively. The surface morphology of the as-grown Si layer was investigated by atomic force microscopy and compositions and layer thicknesses were evaluated by X-ray reflectance, as can be seen in detail in Supplementary Fig. 12.

## Characterization of the Si/SiGe heterostructure
X-ray reflectivity was conducted on a Bruker D8 Discover X-ray diffractometer, equipped with a Cu K$\alpha$ anode, operating at a voltage of 40 kV and current of 40 mA. The X-ray reflectivity spectrum was acquired at ambient temperature with an angular range in $2\theta$ of -0.5° to 8°. The step size was at 0.004°, with a dwell time per step of 30 s. The experimental data was modeled using the pyXRR software[66], which utilizes Fresnel equations for simulating reflectivity and the least-squares minimization approach for curve fitting. The data together with the fit is shown in Supplementary Fig. 12a. The agreement is excellent. A six-layer model was applied, consisting of a SiGe virtual substrate, a layer for damaged SiGe for chemo-mechanical polishing, and the MBE-grown heterostructure (SiGe/Si) with a thin SiO$_2$ layer and air. The best-fit data is described in Supplementary Table 2. Furthermore, the surface topography of the heterostructure was investigated by atomic force microscopy (AFM), using peak force mode. The strained $^{28}$Si surface was homogeneous and smooth with a rms roughness of 140 pm.

## Evaluation of strain of strained Si
To determine the strain value of the strained $^{28}$Si thin film, we used the Raman strain shift coefficient reported for biaxial strain by Wong et al.[31], i.e., 784 ± 4 cm$^{-1}$, taking the Si-Si Raman peak of natural Si at 520.7 cm$^{-1}$ as reference. However, in our case, the strained Si thin film is composed of isotopically pure $^{28}$Si, whose (unstrained) Si-Si Raman mode shifts by approximately +1 cm$^{-1}$ ref. [67] compared to native Si. For this reason, when calculating the strain from Raman spectra, we used 521.7 cm$^{-1}$ as the reference value for the Si-Si Raman peak of strained $^{28}$Si.

## LaNiO$_3$ on LaAlO$_3$ thin film growth and characterization
The LaNiO$_3$ films were grown on a single crystalline LaAlO$_3$ (001)$_{pc}$ substrate (CrysTec GmbH) by pulsed laser deposition using a 248 nm KrF excimer laser (LPXpro, Coherent Ltd.). The films were grown at a substrate temperature of 700 °C under an oxygen partial pressure of 0.1 mbar. The laser fluence was set to 1 J cm$^{-2}$ with a repetition rate of 2 Hz. X-ray diffraction measurements were performed on a four-circle thin-film diffractometer (PanAlytical X'Pert3 MRD) with Cu K$\alpha$1 radiation $\lambda$ = 1.54 Å). X-ray reflectometry was performed to quantify the LaNiO$_3$ film thicknesses, see Supplementary Fig. 13. Surface topography measurements were conducted using atomic force microscopy in a Bruker Multimode 8 scanning probe microscope with Pt-coated Si tips (MikroMasch, $k$ = 5.4 N m$^{-1}$).

## Data availability
The data that support the findings of this study are available from the corresponding author upon request.

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

## Acknowledgements
R.M.W. thanks the Binnig and Rohrer Nanotechnology Center in Rüschlikon/Switzerland and ETH Zürich, Department of Materials. G.K. was funded by the Deutsche Forschungsgemeinschaft (DFG, German Research Foundation) - Project-ID 182087777 - SFB 951. S.H. acknowledges funding from the Deutsche Forschungsgemeinschaft (DFG) under the Emmy Noether Initiative (Project-ID 433878606) and financial support by ETH Zürich Career Seed Grant SEED-16 17-1. M.T. acknowledges the financial support by the Swiss National Science Foundation under project No. 200021_188414. M.F. and L.N. acknowledge the financial support by the Swiss National Science Foundation under project No. 200020_192362. M.C.W. and T.P. are grateful for financial support from the Région des Pays de la Loire under the Etoile Montante Initiative (2022_11808) and the PULSAR Academy. The authors acknowledge the use of the facilities at the Scientific Center for Optical and Electron Microscopy (ScopeM) at ETH Zürich. P.M. acknowledges the financial support by the Einstein International Postdoctoral Fellows program (IPF-2022-727). K-P.G., M.O. and C-H.L. acknowledge the financial support by the Leibniz Association in the project SiGe Quant (project number K124/2018). The authors thank Frédéric Amiard and the vibrational spectroscopy platform at the IMMM, CNRS UMR 6283, Le Mans University as well as A. Tsarapkin and K. Höflich for their support in membrane fabrication and characterisation. We acknowledge support by the Open Access Publication Fund of Humboldt-Universität zu Berlin.

## Author contributions
M.C.W., M.F., R.M.W., and S.H. conceived the project with co-initiation by M.G., R.M.W., K.P.S., and S.M.K. fabricated the nanoporous membranes and graphene samples, and performed the transfers of PAuM onto target substrates, supervised by J.V., M.P., and S.H measured the optical transmission of the PAuM. S.H. performed and analysed the wavelength-dependent Raman measurements. E.B. and S.H. performed the Raman measurements at cryogenic temperatures. G.K. performed the numerical simulations. M.F.S. and M.T. fabricated and characterized the LaNiO$_3$ thin film on LaAlO$_3$. M.C.W. and T.P. performed and analyzed the Raman measurements on the LaNiO$_3$ thin films with support from S.H. L.H., G.M., M.P., M.F., and L.N. assisted in the Raman measurements and numerical simulations. K-P.G., M.O., and C-H.L. fabricated the strained Si thin film samples. P.M. performed and analysed the Raman measurements of the strained Si thin film. R.M.W., G.K., M.F., M.C.W., and S.H. interpreted the results and co-wrote the manuscript with input from all authors. S.H. coordinated and supervised the project.

## Funding

## Competing interests
R.M.W. holds a patent on manufacturing porous gold membranes (Patent US17/294748). Apart from that, the authors declare no competing interests.
