## [Peer Review File · Nature Communications]

Bulk-suppressed and surface-sensitive Raman scattering by transferable plasmonic membranes with irregular slot-shaped nanoporesEditorial Note: Parts of this Peer Review File have been redacted as indicated to maintain the confidentiality of unpublished data.

Reviewer #1 (Remarks to the Author):

Comments:

In this work, the authors reported surface-sensitive and bulk suppressed Raman scattering by porous Au membrane (PAuM). By using graphene as model surface, the authors demonstrated that the surface enhancement and bulk suppression of Raman increased the surface-to-bulk Raman signal ratio by three orders of magnitude. It also demonstrated that 90% of the Raman enhancement occurs within the top 2.5nm of the material. The authors further applied the PAuM to probing structural properties of LaNiO₃ thin film on LaAlO₃. In the presence of the PAuM, the thin film signal is strongly enhanced with respect to the LaAlO₃ substrate signal and Eg-mode splitting in was measured. Generally, the surface enhancement and bulk suppression in Raman scattering by PAuM has been clearly demonstrated. The proposed method may provide a promising way for measurement of surfaces. However, the work only measured the graphene membrane and LaNiO₃ film. In the measurement of LaNiO₃ thin film, the PAuM suppressed both the surface film and bulk substrate. The chosen graphene as model surface is not universal due to the relatively simple Raman peaks. I suggest more tests of other materials be performed to support its potential publication in a high-impact journal.

My questions and comments:

(1) The transmission spectra shown in Fig.1(c) show that the transmittance is over 20% in an arrange of 450-700 nm with maximum (45%) at around 550 nm. This means that the light will be filtered out or reduced significantly out of the range 450-700 nm. The Raman scattering light from surface underneath the PAuM may contain light with different energies. I guess part of the Raman scattering signals may not be detected. How to make sure all the signals can be detected through the PAuM? In addition, the transmittance from plasmonic subwavelength holes is dependent on the shape, size and periodic orders of the holes. Then how to make sure the fabricated plasmonic holes are suitable for surface detection with Raman?

(2) The authors stated that the slot-like nanopores in the membrane act as plasmonic slot antennas and enhance the Raman response of the surface or thin film underneath. There are many types of plasmonic antennas used for Raman enhancement. Did the author test whether other plasmonic antennas, such as bowtie and nanodiscs? Did similar effects of surface enhancement of surface and suppression of bulk in Raman measurement?

(3) The thickness of PAuM used in this work is 20 nm. Why did authors choose this thickness? What about other thicknesses? for example, thicker and thinner than 20 nm?

(4) As shown in Figure 4(b), the Raman signals of substrate LaAlO₃ at regions of 100-160 cm⁻¹ and 450 cm⁻¹-500 cm⁻¹ (Eg) are suppressed when measured with PAuM. However, this measurement also showed that the Raman scattering peaks of LaNiO₃ thin film without PAuM were stronger than those without the PAuM. It seemed that the PAuM did not effectively enhance Raman of the LaNiO₃ thin film.

(5) The authors measured graphene as a model material. The Raman scattering peaks of graphene are relatively simple. I suggest the authors test other materials such as biochemical molecule contaminants on glass or other substrates with PAuM, to test whether surface materials can be effectively detected.

(6) Whether the PAuM transferred to other surfaces can be removed completely without causing damage to the surfaces after the measurement? Have the authors performed such experiments?

(7) Some closely related references are suggested to be discussed and compared with the proposed method, c.f., Opto-Electron Adv 5, 210121 (2022). doi: 10.29026/oea.2022.210121; Opto-Electron Adv 4, 210048 (2021). doi: 10.29026/oea.2021.210048

Reviewer #2 (Remarks to the Author):

The authors have presented an experimental demonstration of the surface-sensitive and bulk-

suppressed Raman spectrum using the transferable porous gold membrane (PAuM). The PAuM, comprising of 'slot-like nanopores', effectively confines the excitation laser to the surface of interest while suppressing laser penetration into the bulk, thereby enhancing the surface-to-bulk signal ratio. The successful validation of the main idea through the analysis of graphene and LaNiO₃ film using PAuM is noteworthy. The authors have made significant progress in achieving SERS surface analysis of materials, which is usually challenging since the SERS hotspots are difficult to confine at the material surface, especially for non-conductive materials. A few issues should be addressed before its publication, as listed below:

1. It is important to determine the surface-to-bulk signal ratio of other commonly used SERS substrates such as Au/Ag nanoparticles, as an additional calibration. These SERS substrates usually can enhance the surface signals, while fail to achieve the significant suppression of bulk signals. Therefore, in addition to comparing PAuM with and without PAuM cases, conducting a comparative study between PAuM and other common SERS substrates would be helpful.
2. The authors have demonstrated the effectiveness of PAuM on flat surfaces/films such as graphene on Si and epitaxially grown LaNiO₃ on a 100-LaAlO₃ substrate. However, it would be beneficial to investigate the applicability of PAuM to rough surfaces, such as the surface of a Li-ion battery electrode. The feasibility of using PAuM on rough surfaces may depend on the flexibility of the PAuM membrane.
3. One drawback of PAuM for surface analysis of materials is the direct contact between the Au membrane and the material surface, which may lead to charge transfer as well as hot carrier injections between PAuM and the materials. It is essential to address this concern and discuss potential methods to mitigate or overcome these issues.
4. Another drawback of PAuM is that the average surface enhancement factor of Raman signal is relatively low (on the order of 100). It would be valuable for the authors to discuss potential strategies for further improving the enhancement factor.
5. In Table I, it is observed that the G-mode and 2D-mode peaks of graphene exhibit different enhancement factors (EFs) when excited by 660 nm and 785 nm lasers. The EF of the G-mode peak is slightly higher than that of the 2D-mode peak for 660 nm excitation, whereas the EF of the 2D-mode peak exceeds that of the G-mode peak by five times for 785 nm excitation. Further discussion is needed to explain the difference in EFs between the G- and 2D-modes.
6. Disordered nanoholes with various sizes often result in broadband absorption and enhancement. However, the PAuM in this study only exhibits several separate peaks in the range of 650 nm to 850 nm. Considering the benefits of broadband enhancement in arbitrary conditions, it would be interesting to explore if the response bandwidth of PAuM can be improved by optimizing the manufacturing process.
7. The position of the edge of the PAuM should be indicated in Fig. 2j, and the scan line of Fig. 2j should be marked in Fig. 2f.
8. The dielectric constant of the materials used in the simulation should be provided.
9. There are a few typos in the manuscript, e.g. Page 6 Fig. 1c, the unit of wavelength should be labeled. Page 7 "the energy of the plasmonic resonance depends one " ; Page 8 "Note the 2D-mode intensity for bare graphene and 785 nm is below the noise level" should be "Note that the 2D-mode intensity of bare graphene for 785 nm is below the noise level."? Page 10, ".the Raman scattered light for a non-porous membrane of comparable thickness, see Fig.2(c).. " should be Fig.1(c)?

Reviewer #3 (Remarks to the Author):

Reviewer #4 (Remarks to the Author):

In this report, authors reported on the results of reducing the bulk Raman signal and increasing only the surface Raman signal by using PAuM (porous gold membrane).

As a result of measuring Raman scattering by placing the PAuM developed by the authors on a graphene/Si sample, it was confirmed that the Raman scattering on the surface was greatly enhanced. Authors explained that the PAuM structure acts as a nano slot antenna, increasing the Raman signals from the surface and blocking Raman signals from the bulk.

Based on their experimental results, it was experimentally proven that bulk Raman scattering can be efficiently removed and only surface Raman scattering can be measured through nano slot metallic structures.

Their results are believed to suggest a new application field for porous metal materials, that is, a method to effectively separate surface and bulk Raman signals and measure only surface Raman scattering. Therefore, I expect to receive widespread attention in the fields of Raman scattering society and plasmonics field because their suggested new methodology can achieve great effect compared to the existing Raman spectroscopy.

But, I have some questions on their report, and I hope to hear from them about those.

1. The porous membrane presented by the authors does not seem to be able to control the polarization and desired position.

In order to be useful as an actual measurement method, it seems necessary to adjust the shape and direction of the nano slot to obtain more information using incident polarization or to obtain information at a desired location on the sample.

It seems good to mention the authors' alternative methods in the revised report.

2. Rather than using the porous membrane structure suggested by the authors as the above method, it seems to be more useful in various applications to directly manufacture the desired slot on a thin metal film.

It seems necessary to describe whether it is possible to reduce the bulk Raman signal and obtain only the surface Raman signal.

3. A slot that is too wide will not be able to block the bulk Raman signal and is expected to have no plasmonic field enhancement effect.

Therefore, there appears to be a boundary value for the slot width at which surface enhancement/bulk depression occurs suggested by the authors.

It seems necessary to present the authors' opinions on this.

4. The depression of the bulk signal due to the interference between SiO₂ and Si is expected to repeatedly show maximum and minimum values depending on SiO₂ film thickness.

Therefore, I think that the thickness of SiO₂ thin film on substrate has optimal thickness for the suppression.

I think that it's better to comment about that in the revised version.

5. It looks good to express the lines in Figure 4 with different colors.

Reviewer #5 (Remarks to the Author):

In this work, the authors report on a new technique to enhance the Raman signal from the surface of the sample selectively. Since surface-specific information is needed in many studies, this technique has a potential for an impact in the field. The manuscript is written in a clear and straightforward manner, and the presentation is mostly reasonable. There are a few points that need to be addressed before this work can be published.

1. Because PAuM has to be transferred onto the surface of the sample, the contact between the sample and the PAuM should be very good. Is there any way to check this? Does it mean that this method can be used only on samples with a very smooth surface?
2. In the Raman spectroscopy of graphene, the SiO₂ thickness is a critical parameter. Although the value is given in Methods, it would be better to specify it when it first appears in the main text.
3. Can the authors plot the spectra in Figures 2 b-d in the unit of counts per second (CPS)? This would allow the readers to judge the actual intensity level of the signal.
4. If the Raman signal is mostly from the top surface layer in the case of LNO on LAO, the intensity and the line shape should be independent of the thickness of the film. This is an easy check and would be strong evidence that supports the authors' claim. If the authors did not do it, it is strongly advised.
5. The authors claim that the ratio between surface enhancement and bulk suppression can be tuned by changing the PAuM thickness. Data to support this claim are needed.
6. According to Ref. #36, the enhancement factor for a SiO₂ thickness of 30 nm is almost 0 regardless of the wavelength. However, in Table S1 (incorrectly referred to as Table S3 in the text), the numbers are substantial. The authors are advised to check the numbers.

In the discussion section, it is advisable to add some comments on the limitations of this technique. Some of the issues are: (1) This is essentially a destructive method. The sample cannot be used for other measurements after the PAuM is transferred. (2) Since the pore shapes and orientations are random, polarization-dependent measurements cannot be performed. (3) Since the resonance is wavelength-specific, excitation-wavelength-dependent properties cannot be studied.

Response to Reviewers 23-42440

Reviewer 1

In this work, the authors reported surface-sensitive and bulk suppressed Raman scattering by porous Au membrane (PAuM). By using graphene as model surface, the authors demonstrated that the surface enhancement and bulk suppression of Raman increased the surface-to-bulk Raman signal ratio by three orders of magnitude. It also demonstrated that 90% of the Raman enhancement occurs within the top 2.5nm of the material. The authors further applied the PAuM to probing structural properties of LaNiO₃ thin film on LaAlO₃. In the presence of the PAuM, the thin film signal is strongly enhanced with respect to the LaAlO₃ substrate signal and E_g-mode splitting in was measured. Generally, the surface enhancement and bulk suppression in Raman scattering by PAuM has been clearly demonstrated. The proposed method may provide a promising way for measurement of surfaces. However, the work only measured the graphene membrane and LaNiO₃ film. In the measurement of LaNiO₃ thin film, the PAuM suppressed both the surface film and bulk substrate. The chosen graphene as model surface is not universal due to the relatively simple Raman peaks. I suggest more tests of other materials be performed to support its potential publication in a high-impact journal.

We thank the reviewer for the evaluation of the manuscript. Based on some of the reviewer's comments above we believe that several aspects of our work have not been described with sufficient clarity. We apologize for this misunderstanding and address these aspects and all other concerns in the following and/or in detail after each comment/section.

Reviewer 1 states that *"the chosen graphene as model surface is not universal due to the relatively simple Raman peaks"*. This comments suggests the need to clarify the role of graphene as a model surface and why it is in fact very close to universal:

- 1. The Raman signals of graphene can be detected without any enhancement.** Thanks to its single-layer character, graphene acts as "surface-only material" and a disturbing bulk contribution to the Raman signal is absent. As the Raman signals of graphene can be detected without any enhancement, we can quantify the surface enhancement provided by our porous membrane through comparison to the non-enhanced graphene Raman-spectrum.
- 2. The Raman intensity of graphene is independent of wavelength and polarization.** The Raman intensity of graphene is independent of wavelength and polarization. Therefore, any Raman intensity change is solely due to the membranes. Furthermore, this enables us to relate the Raman signal amplification of different wavelengths directly to each other.
- 3. The simplicity of the graphene Raman peaks is ideal to demonstrate the effect of surface-enhancement proposed here.** Since the Raman peaks of graphene are very well understood, they are ideal to verify that the presence of the membranes does not induce major mechanical or other structural modification of the surface such as strain, compression or defects.

In summary, graphene is the ideal model surface to showcase the core features of our approach.

Regarding our measurements on the LaNiO₃ thin film on LaAlO₃ substrate, the reviewer comments that “*in the presence of the PAuM, the thin film signal is strongly enhanced with respect to the LaAlO₃ substrate signal and Eg-mode splitting in was measured*” and that “*In the measurement of LaNiO₃ thin film, the PAuM suppressed both the surface film and bulk substrate.*” Reviewer 1 further states that “*the work only measured the graphene membrane and LaNiO₃ film*”. Based on these comments, we realized that we failed to describe our measurements of the LaNiO₃/LaAlO₃ structure with sufficient clarity.

Importantly, the PAuM-enhanced Raman signal does not represent an average of the vibrational signal of the entire LaNiO₃ film. But, the PAuM-enhanced Raman signal stems primarily from **the surface layer**. The Raman signal of the entire LaNiO₃ thin film can be measured with conventional Raman spectroscopy without requiring our PAuM. Using the PAuM, on the other hand, we have revealed an additional Raman feature (peak splitting) that arises from the **surface of the LaNiO₃ thin film** (see also our response to Question 4 of Reviewer 1). This feature does not appear in the Raman response of the LaNiO₃ thin film itself without PAuM.

The ability itself to access the structural information of the surface layer of a 20 nm thin film offers, in our opinion, an entirely new perspective for the understanding of physical phenomena in complex oxides. We hope that clarifying this misunderstanding will convince the reviewer that our method is indeed surface-sensitive and merits publication in Nature Communications.

Further, **we implemented the suggestion of Reviewer 1 to test other materials** to support the potential of surface-sensitive Raman scattering enabled by PAuM. We chose to quantify the strain in a strained 12.5nm thick quantum well layer of isotope-enriched ²⁸Silicon that is part of a Si/Ge heterostructures, see figure to the left. Such heterostructures are utilized as electron spin qubits in gate-defined silicon quantum dots and represent a promising and fast-developing technology for quantum computing.

The strain is used to lift the conduction band valley degeneracy which is a prerequisite for obtaining electron-spin qubits. It is therefore of high importance to quantify the strain in the quantum well layer. Obtaining a Raman signal of the thin film requires a dedicated UV Raman spectroscopy setup with a shallow penetration depth into the material.

Using PAuM placed on top of the 12.5nm thick strained Si layer, we observe the Raman signal of the thin Si film at 514cm⁻¹ for 638nm, and more pronounced at 785 nm excitation. The bulk Raman signals of the SiGe layer and bulk Si are partially suppressed by the PAuM, which further improves the detectability of the thin film's Raman signal. From the Raman shift, we extract a biaxial strain of 0.96%, in very good agreement with XRD data (1.01%). Without PAuM, the Raman signal of the thin film is not detected.

This further test of an alternative material unequivocally demonstrates the benefit of surface-sensitive and bulk-suppressed Raman spectroscopy enabled by PAuM, underscores its versatility, and strongly supports publication in Nature Communications.

Changes to the manuscript:

We have clarified the part of the LaNiO₃ section and added the measurement of the thin film strained silicon in the main text of the manuscript as a second application case for our approach.

My questions and comments:

Reviewer 1, Q1: The transmission spectra shown in Fig.1(c) show that the transmittance is over 20% in an arrange of 450-700 nm with maximum (45%) at around 550 nm. This means that the light will be filtered out or reduced significantly out of the range 450-700 nm. The Raman scattering light from surface underneath the PAuM may contain light with different energies. I guess part of the Raman scattering signals may not be detected. How to make sure all the signals can be detected through the PAuM? In addition, the transmittance from plasmonic subwavelength holes is dependent on the shape, size and periodic orders of the holes. Then how to make sure the fabricated plasmonic holes are suitable for surface detection with Raman?

Our answer: The transmittance is not essential for Raman signal detection, as the enhanced Raman signals are generated within the pores and are thus not transmitted through the porous gold membrane. Signals stemming from directly underneath the gold membrane, at a position without pore, will be very weak as (i) minimal excitation light will reach this part and (ii) hardly any Raman signal can make its way through the membrane. Therefore, the low transmission is actually a desired effect and enables a strong suppression of the bulk contribution to the Raman signal. In contrast, within pores we gain high field intensities due to plasmonic resonances (equivalent to those around nanoparticles - Babinet's principle). Thus, with respect to Raman scattering sites (pores) the low transmission is no problem.

The second part of the reviewer's question addresses the shape, size and order of the pores. Indeed, these parameters are important to generate hotspots. Our essentially *random* pore morphologies ensure resonant pores at basically any wavelength. Of course, in practice the fabrication process (details below and in SI) effectively yields a non-uniform distribution of resonances over the visible spectral range, which can be seen in Fig1c (more resonances around 700 nm - 800 nm). The distribution of resonances can be controlled by the fabrication parameters (i.e. gold thickness).

Maybe even more important than the fact that the membrane is always resonant at some location, individual plasmonic resonances are very broad. Thus, if an excitation laser is resonant to a pore, the corresponding Raman lines will also be resonant, yielding the well-known E^4 enhancement. Thus, in practice one can almost choose any desired laser wavelength for excitation, though preferentially a laser wavelength at around 700 nm - 800 nm in the current example of Fig1c.

Reviewer 1, Q2: The authors stated that the slot-like nanopores in the membrane act as plasmonic slot antennas and enhance the Raman response of the surface or thin film underneath. There are many types of plasmonic antennas used for Raman enhancement. Did the author test whether other plasmonic antennas, such as bowtie and nanodiscs? Did similar effects of surface enhancement of surface and suppression of bulk in Raman measurement ?

Our answer: Reviewer 1 is right in stating that the enhancement of surface Raman signals is not unique to plasmonic slot antennas and that it should also occur for other antenna types. We have not tested other structures since our focus is on demonstrating the concept of surface-sensitive and bulk-suppressed Raman spectroscopy by PAuM. This approach is not unique to our membranes - see discussion in the paper - but our membranes are a simple and effective material to demonstrate the concept, its performance, and how it is applied in useful scenarios.

There are innumerable papers on various antenna designs, some of which would yield better Raman enhancements than simple pores, in principle. The only prerequisite for enhancing surface Raman signals is that the enhanced nearfield extends below the antenna structures, which is basically the case for all plasmonic antenna types that have a nearfield hotspot with a corner-type feature close to the surface of interest. This can be seen i.e. in the simulated field intensity of a plasmonic nanodisc dimer in Fig. 5(b) in Heeg et al. Nano letters 13 (1), 301-308 (2013). The same applies to bowtie antennas.

In general, plasmonic antennas like bowtie or nanodisc dimers are unsuitable for bulk suppression compared to plasmonic films. In realistic experimental scenario (633 nm excitation, laser spot diameter 1 μm) a plasmonic film with one single prototypical plasmonic nanopore will transmit around 20% - see main paper Fig. 1(c) - of the laser intensity and bulk Raman by signals, which leads to a combined suppression factor of 50. A single plasmonic dimer, on the other hand, absorbs around 1% - 2.5% of the laser intensity and bulk Raman signals when in resonance (data from Rangacharya et al. ACS Sens. 2020, 5, 7, 2067–2075). Ignoring scattering for simplicity, this yields 97.5% transmission both ways. We obtain a suppression factor of 1.05 which is clearly much lower than for the plasmonic film. Furthermore, a dimer type structure only contributes to bulk Raman signal suppression for light that couples to its resonant excitation, which is polarization selective.

Overall, the best tool to realize surface-enhanced and bulk-suppressed Raman scattering is a gold film for efficient bulk suppression and inverse plasmonic antennae for surface-enhancement. In principle, any type of inverse antenna design - following Babinet's principle - is suitable. Designs like inverse bowtie antennas, however, require sophisticated fabrication such as electron beam lithography or Focus ion beam milling. On top, each antenna has to be written or milled individually. Broadband surface-enhancement also

requires sets of antennae with different dimensions. On top, transferring these antennas is also a challenge. Our nanoporous membranes, on the other hand, combine the advantages of being easy to fabricate at wafer scale and that they can be readily transferred onto the target surface. The randomness of the pore shape and design also ensured surface-enhancement over a broad excitation wavelength range.

Given the advantages of our nanoporous membranes and the complexity associated with fabricating and transferring Au films with well-defined inverse plasmonic antennae, and considering the proof-of-principle nature of our work, we have not performed such comparative studies.

Changes to the manuscript

We have added a paragraph on alternative structures to achieve surface-sensitive and bulk suppressed Raman spectroscopy in the Discussion section of the paper.

Reviewer 1, Q3: The thickness of PAuM used in this work is 20 nm. Why did authors choose this thickness? What about other thicknesses? for example, thicker and thinner than 20 nm?

Our answer: We chose a thickness of 20 nm due to the favorable morphology of the PAuM at this thickness: Any thickness above 30 nm is unsuitable as the number of pores is vanishingly small. For membranes thinner than 15 nm, the bulk suppression will be very weak as the percolation network is only just forming. The morphologies of the different membranes can be found in Wyss et al. ACS Applied Materials & Interfaces 14(14, 16558-16567 (2022), Fig. S3 in the supplementary information. We have added the figure below:

Figure: Freestanding PAuM of thicknesses 5 nm to 25 nm in 5 nm increments from SI of Wyss et al. ACS Applied Materials & Interfaces 14(14), 16558-16567 (2022), Fig. S3. The number and size of pores reduces with increasing thickness, while narrow, slot-like pores - essential for hotspot formation - are most prominent for 20 nm and 25 nm thick PAuM. We therefore chose a thickness of 20 nm as a compromise between number of pores and shape/form. Scale bars from left to right: 20 μm , 1 μm and 300 nm. Reprinted with permission from Wyss, Roman M., et al. "Freestanding and permeable nanoporous gold membranes for surface-enhanced Raman scattering." ACS Applied Materials & Interfaces 14(14), 16558-16567 (2022). Copyright 2022 American Chemical Society.

Reviewer 1, Q4: As shown in Figure 4(b), the Raman signals of substrate LaAlO₃ at regions of 100-160 cm⁻¹ and 450 cm⁻¹-500 cm⁻¹ (E_g) are suppressed when measured with PAuM. However, this measurement also showed that the Raman scattering peaks of LaNiO₃ thin film without PAuM were stronger than those without the PAuM. It seemed that the PAuM did not effectively enhance Raman of the LaNiO₃ thin film.

Our answer: Indeed, the overall signal stemming from the LaNiO₃ film appears weaker with membrane than without membrane. However, the spectra with and without membrane yield different information.

Without membrane: bulk and surface of the LaNiO₃ thin film is probed and its signal is dominated by the bulk contribution of the LaNiO₃ film.

With membrane: Thanks to the strong light absorption of the membrane, the accessible scattering volume is in first approximation reduced to the region of the pores. At the position of the antenna-like pores, the signal of the surface of the LaNiO₃ thin film is enhanced rather than the entire film. This is shown by the mode splitting, which is absent in the bulk of the thin film. Therefore, the signal of the LaNiO₃ surface is enhanced and becomes visible. The signal of the bulk of the LaNiO₃ film, however, is suppressed due to the reduced Raman scattering volume.

Therefore, the overall signal is reduced while only the surface signal is enhanced that remains obscured without PAuM.

Reviewer 1, Q5: The authors measured graphene as a model material. The Raman scattering peaks of graphene are relatively simple. I suggest the authors test other materials such as biochemical molecule contaminants on glass or other substrates with PAuM, to test whether surface materials can be effectively detected.

Our answer: We thank the reviewer for this suggestion. We chose graphene on SiO₂ as a model material precisely because of its relatively simple Raman spectrum, which is furthermore independent of excitation wavelength and polarization. These properties make it the most suitable material to investigate the effect, as shown for different wavelengths (shown in Figure 2) and the depth dependence of the enhancement (Figure 3). Graphene as a model surface has been introduced and extensively studied (see i.e. Schedin, F, ACS nano 4.10 (2010): 5617-5626. Heeg, S et al. Nano letters 13.1 (2013), Heeg, S, et al. physica status solidi (RRL)—Rapid Research Letters 7.12 (2013), Zhang, S. G.. Applied Physics Letters 104.12 (2014). Wasserth, S. et al. Physical Review B 97.15 (2018))

As an additional use case of our membranes beyond the LaNiO₃, we therefore demonstrate the application of the technique to a strained Si thin film. We are able to resolve specific Raman features using our PAuM approach, which could not be resolved otherwise as the bulk contribution would obscure its surface signature.

Furthermore, while our PAuM technique has many more application cases, it is most useful for investigation of materials where a specific surface signature needs to be detected, which is otherwise obscured by the underlying bulk. For molecules on substrates where both materials have sufficiently distinct Raman signatures, our PAuM have a rather incremental benefit. We also want to mention that the detection of molecules can be done by SERS, with substrates that enable even single molecule detection.

Changes to the manuscript

Following the request of Reviewer 1 to test other materials, we used our PAuM to extract the strain in a strained 12.5 nm thick Silicon layer that is part of a Si/Ge heterostructure. These structures are a promising material for quantum information processing. We observe the Raman signal of the strained silicon when covered with our PAuM. This signal is not detectable without the PAuM. This example further elucidates the applicability of PAuM to study surfaces and thin films.

Reviewer 1, Q6: Whether the PAuM transferred to other surfaces can be removed completely without causing damage to the surfaces after the measurement? Have the authors performed such experiments?

Our answer: The gold membranes can be removed quite easily from many materials, e.g. using a potassium-iodide (KI) gold etch, if the substrate material sustains such a treatment. Furthermore, simple peeling off of the membrane can also be performed, especially for ceramics, where gold typically has a very low adhesion. Another method is simple scratching of the membrane (see image below).

In the image above, the gold membrane was found peeled after transferring (left). Measurement of the Raman signal (right) unveiled, that the 14 nm LaNiO₃ film was not damaged by the gold membrane as its spectrum is in very good agreement to Raman spectra of same thickness found in literature (see for example PRB 94, 014118 (2016)).

While in this work membranes of up to 1 cm x 1 cm were transferred - effectively covering the entire sample surface - it would also be possible to transfer very small PAuM (100 μm x 100 μm). Then, the surface under investigation is not entirely covered and the sample can be used in further experiments or processing.

Changes to Manuscript:

We have added a paragraph in the Supplementary Information Section 1 mentioning removal of PAuM from substrates by etching or scratching.

Reviewer 1, Q7: Some closely related references are suggested to be discussed and compared with the proposed method, c.f., Opto-Electron Adv 5, 210121 (2022). doi: 10.29026/oea.2022.210121; Opto-Electron Adv 4, 210048 (2021). doi: 10.29026/oea.2021.210048

Our answer: We thank the reviewer for bringing our attention to these papers. We have added and discussed them in the introduction.

Reviewer #2 (Remarks to the Author):

The authors have presented an experimental demonstration of the surface-sensitive and bulk-suppressed Raman spectrum using the transferable porous gold membrane (PAuM). The PAuM, comprising of 'slot-like nanopores', effectively confines the excitation laser to the surface of interest while suppressing laser penetration into the bulk, thereby enhancing the surface-to-bulk signal ratio. The successful validation of the main idea through the analysis of graphene and LaNiO₃ film using PAuM is noteworthy. The authors have made significant progress in achieving SERS surface analysis of materials, which is usually challenging since the SERS hotspots are difficult to confine at the material surface, especially for non-conductive materials. A few issues should be addressed before its publication, as listed below:

We thank the reviewer for the positive feedback. We have addressed all comments and suggestions below.

Reviewer 2, Q1: It is important to determine the surface-to-bulk signal ratio of other commonly used SERS substrates such as Au/Ag nanoparticles, as an additional calibration. These SERS substrates usually can enhance the surface signals, while fail to achieve the significant suppression of bulk signals. Therefore, in addition to comparing PAuM with and without PAuM cases, conducting a comparative study between PAuM and other common SERS substrates would be helpful.

Our answer: Separated plasmonic antennas are a rather impracticable choice if one aims for a structure allowing bulk suppression. We refrain here from performing more numerical simulations of such nanoparticle-based structures as it is hardly possible to draw a fair comparison between PAuM and these structures. Any plasmonic nanoparticle can in principle support similarly enhanced nearfields at the interface of the particle to substrate. Spatial overlap with surface as well as intensity of the nearfields will depend on the shape of the particles (e.g. radius of curvature). Furthermore, factors such as packing density has a strong influence as well, as studied before (see Solís, Diego M., et al. ACS photonics 4.2 (2017): 329-337.). A random, but densely packed arrangement of flat silver particles might be best for surface excitation including some degree of bulk suppression, however, still particles will be prone to heating much more than the PAuM that can dissipate away heat quite effectively. Also, dense packing of nanoparticles can be cumbersome to control, e.g., by spin-casting with many runs to optimize for every new substrate due to different wetting conditions of solvents. Further, in practice, it turns out that particle-based SERS approaches need a thermal treatment to around 300°C (see Lu et al. Carbon 86 (2015), 78-85) as only then particles form sharp edges at the interface to the substrate yielding plasmonic hot spots where they are needed. This heat treatment needs to be adjusted for different substrates and comes with the danger to harm the substrate. A last argument against particle-based approaches is the generation of hot-electrons that might alter the interface under study. Hot-electron generation is expected to be significantly enhanced for small particles as compared to our smooth membrane (see comment to your Q3).

In conclusion, a "fair" comparison is hardly possible and an unfair comparison is of little value for the reader. Even ignoring some of the above mentioned disadvantages of particles,

we would have to invent a new particle-based structure to compete with the PAuM in terms of bulk suppression and enhancement factor. Such a comparative study requires significant experimental effort or extended numerical simulations, which is far beyond the scope of this work.

Changes to Manuscript:

However, we agree that it is helpful to highlight the benefits of the PAuM by comparing it qualitatively to alternative structures. We have added a general discussion to the main manuscript (last three paragraphs before the chapter conclusions) on various aspects raised by the Reviewers.

Reviewer 2, Q2: The authors have demonstrated the effectiveness of PAuM on flat surfaces/films such as graphene on Si and epitaxially grown LaNiO₃ on a 100-LaAlO₃ substrate. However, it would be beneficial to investigate the applicability of PAuM to rough surfaces, such as the surface of a Li-ion battery electrode. The feasibility of using PAuM on rough surfaces may depend on the flexibility of the PAuM membrane.

Our answer: We thank the reviewer for this comment as the investigation of battery materials is indeed an interesting use case. At this point, however, this would stretch beyond the scope of this work and is therefore left for future studies. PAuM are stable enough to span over holes of several μm diameter, essentially forming a drum-like structure, without breaking. Furthermore, they are flexible on a macroscopic scale, see e.g. the transferred PAuM on polycarbonate track-etched membranes in the SI of the paper Wyss et al. ACS Applied Materials & Interfaces 14(14, 16558-16567 (2022)). That membrane was transferred by heat-induced bonding and peeling the PCTE-PAuM from the support wafer.

Depending on the specifics of the roughness of the surface, the PAuM would be in contact with that surface at a specific number of positions per surface area. This corresponding contact area A_c might be described with some ratio r to the overall area A_{tot} like $r=A_c/A_{\text{tot}}$. Then the likelihood of detecting a bulk-suppressed Raman signal per illuminated and observed surface area is simply weighted with this ratio r as most of the Raman signal from the interface stem from the regions where the near field overlaps with the surface.

We would like to point out that we chose the materials (graphene, LaNiO₃) specifically due to their Raman spectroscopic properties i.e. well understood Raman spectra (Graphene), strain quantified by a complementary method (XRD) for the strain Si-layer, and expected observation of a mode splitting for the surface (LaNiO₃). Furthermore, the transfer of PAuM on these materials is straight-forward and can be achieved using standard 2D material transfer processes (Wyss et al. ACS Applied Materials & Interfaces 14(14, 16558-16567 (2022))). The transfer to rough surfaces would need process development and optimization to achieve optimal results, which is in principle possible.

Reviewer 2, Q3: One drawback of PAuM for surface analysis of materials is the direct contact between the Au membrane and the material surface, which may lead to charge transfer as well as hot carrier injections between PAuM and the materials. It is

essential to address this concern and discuss potential methods to mitigate or overcome these issues.

Our answer: We agree that this should be addressed. Literature states that hot electrons are best generated in extreme hot spots (e.g., dimer antennas), in relatively small particles or on rough surfaces (see Y. J. Jang, K. Chung, J. S. Lee, C. H. Choi, J. W. Lim, D. H. Kim, ACS Photonics 2018, 5, 12 4711; A. O. Govorov, H. Zhang, Y. K. Gun'ko, The Journal of Physical Chemistry C 2013, 117, 32 16616; J. B. Khurgin, Nanophotonics 2020, 9, 2 453). In many situations this is also where Raman enhancement occurs. Conclusively Raman signatures have been probed to understand in particular hot electron transfer processes to nearby materials. See e.g. the review by Yang et al. Cell Reports Physical Science 1.9 (2020).

However, as PAuMs do not fulfill the above requirements, we estimate that a PAuM is an inefficient structure to generate hot electrons and also that transfer to nearby materials is weak. The PAuM-to-substrate interface is very smooth as PAuMs are fabricated on flat SiO₂ substrates similar to template stripping samples. Instead of small particles the PAuM is an extended structure and the generated hotspots in pores feature moderate enhancements when compared to, e.g., tiny gaps in dimer or nanoparticle on mirror antennas.

Another well studied aspect is the distance dependence of hot electron transfer. The efficiency drops quickly and depends on the materials composing the interface. The review above mentions that a 1nm barrier suppresses the transfer completely while plasmonic near-fields of the membrane easily extend beyond a 1 nm gap. The transfer efficiency will surely also depend on the barrier height. From this behavior we can conclude that a studied substrate can be protected from potentially unwanted hot electron transfers by a very thin layer, e.g., a hBN layer or a thin polymer coating (e.g. by spincoating a dilute solution) or a treatment of the membrane with molecules that are known to attach to gold efficiently (also used for stabilization of gold nanoparticles).

Concerning charge transfer (without hot electrons) we can state that this is part of the often discussed “chemical enhancement” mechanism of Raman enhancement and thus would not be unwanted, though we believe that “electromagnetic enhancement” will be the relevant mechanism in most of the situations due to its predicted higher enhancement factor.

Changes to manuscript: We have added the following paragraph in the Discussion section of the main manuscript:

“In addition, metallic nanostructures for plasmonic enhancement may also entail the photoinjection of hot electrons into the material. This effect is particularly pronounced for small nanoparticles, rough interfaces and nanogapped structures with extreme field confinement [55–57]. In contrast, our plasmonic membrane is an extended structure which also features a very smooth interface since it was fabricated on flat SiO₂, similar to plasmonic structures obtained by template stripping. We conclude that our membranes will hardly undermine charge carrier momentum conservation which is needed for efficient electron or hole transfer [56, 57]. Nevertheless, the role of hot carrier injection in surface-sensitive Raman scattering by plasmonic membranes and its effect on the Raman

response of the surface or film under study requires further clarification and will be subject of future work. A potential route to prevent hot carrier injection is inserting an insulating barrier, such as a monolayer of hexagonal boron nitride, between the PAuM and the surface [58]

Reviewer 2, Q4: Another drawback of PAuM is that the average surface enhancement factor of Raman signal is relatively low (on the order of 100). It would be valuable for the authors to discuss potential strategies for further improving the enhancement factor.

Our answer: We agree with the reviewer that the enhancement factor of 100 appears low but we argue that this number is not the correct figure of merit here especially when compared to common SERS substrates. The factor 100 is the experimentally observed enhancement based on comparing the integrated Raman intensity arising from a few pores within the laser spot to uncovered graphene, for which the entire illuminated area contributes to the signal. Upon including the pores' area from which the enhancement arises, we obtain local enhancement factors of the order of 10^4 to 10^5 from individual pores, see Wyss et al. ACS Applied Materials & Interfaces 14(14), 16558-16567 (2022).

Both the local and experimentally observed enhancement factors can be increased by designing PAuM with oriented (i.e. parallel) pore of specific size and aspect ratio. One of the following strategies could be employed:

1. FIB patterning: Deposition of a continuous gold film of a desired thickness (e.g. by evaporation on an intermediate chrome layer) and subsequent patterning using a focused ion beam system. Pores of desired size, orientation and density can be readily achieved. The disadvantage of this method is the rel. slow and expensive processing.
2. E-beam lithography: Using e-beam, photoresist structures are defined on a carrier wafer where pores should be created in the membrane. Gold is subsequently evaporated on the carrier wafer, where a following lift-off process removes the predefined e-beam structures including gold on those structures. On the carrier, the gold membrane with desired pores is left behind.
3. Patterned hard mask: Patterning the carrier wafer (e-beam/RIE etching, FIB, etc.) with pores of desired aspect ratio and density yields a "hard mask" with holes. If gold is evaporated at an angle at these wafers, the holes do not/only partially fill, creating a porous gold membrane with desired structures.

Further strategies to obtain higher enhancement factors include nanoimprint lithography and block copolymers. All the methods above, however, have in common that they require a significantly higher effort with respect to both, working hours processes and equipment, as compared to our nanoporous membranes, which are basically fabricated in one step on wafer scale.

[Redacted]

In summary, more elaborate fabrication techniques will yield higher surface Raman signal enhancement, but lack the simplicity, scalability and broad-band applicability of our nanoporous membranes.

Changes to Manuscript:

Based on this question, we have added a paragraph in the discussion mentioning methods such as FIB, photolithography or nanoimprint lithography to create ordered, well-defined arrays of pores, which would lead to a stronger enhancement of the Raman signal compared to PAuM at the cost of more laborious, complex and time-consuming manufacturing.

“In general, surface sensitive and bulk suppressed Raman spectroscopy is not limited to PAuM. The defining features of a plasmonic structure to enable surface sensitive Raman scattering is (i) the formation of a hotspot close to the surface, (ii) the suppression of bulk signals and (iii) the ability to sustain laser irradiation without structural damage. While other structures such as nanoparticle films or isolated bowtie structures offer one or two of these defining features, none offer the combination of all three. In contrast, PAuM and other types of plasmonic membranes fulfill these requirements. Membranes with nanopores of defined geometry, for instance, can be tailored to independently maximize bulk suppression and surface enhancement for specific excitation wavelengths. However, manufacturing this type of structures requires dedicated instrumentation (e.g. focused ion beam) often employing time-consuming serial processes, whereby one pore is created at the time, limiting the size of the membrane to a few hundreds of μm^2 . Our PAuM, on the other hand, are manufactured by simple evaporation on wafer-scale and provide broadband surface enhancement without cumbersome and laborious processing steps.”

Reviewer 2, Q5: In Table I, it is observed that the G-mode and 2D-mode peaks of graphene exhibit different enhancement factors (EFs) when excited by 660 nm and 785 nm lasers. The EF of the G-mode peak is slightly higher than that of the 2D-mode peak for 660 nm excitation, whereas the EF of the 2D-mode peak exceeds that of the G-mode peak by five times for 785 nm excitation. Further discussion is needed to explain the difference in EFs between the G- and 2D-modes.

Our answer: This behavior is not unusual in plasmon-enhanced Raman spectroscopy, especially when graphene is used to probe the enhancement, see i.e. Schedin et al. ACS Nano 2010, 4, 10, 5617–5626 or Wyss et al. ACS Applied Materials & Interfaces 14(14), 16558-16567 (2022). Especially the 2D-mode has a high energy (~325 meV) that can be comparable to the width of a typical plasmonic resonance, see Heeg et al. Nano letters 13.1 (2013): 301-308. This means that depending on the exact energy of a nanopore's resonance and its width, the excitation and the 2D mode (or one of the the two) overlaps with the

resonance of a given nanopore. The G-mode has an energy of 200 meV. This means that the G- and 2D mode typically have different enhancement factors at each nanopore.

Since the measured Raman intensity comes from many nanopores, the resulting enhancement factors for the G- and 2D- mode vary at any given location and for different wavelengths. The purpose of using graphene in this work is to estimate the order of magnitude of the Raman enhancement and to identify which excitation wavelength that should be used to probe surface and thin films.

Reviewer 2, Q6: Disordered nanoholes with various sizes often result in broadband absorption and enhancement. However, the PAuM in this study only exhibits several separate peaks in the range of 650 nm to 850 nm. Considering the benefits of broadband enhancement in arbitrary conditions, it would be interesting to explore if the response bandwidth of PAuM can be improved by optimizing the manufacturing process.

Our answer: Manufacturing processes to create PAuM with aligned pores of specific size, aspect ratio and density are possible (see our answer to Q4). These membranes will subsequently also exhibit different absorption characteristics. In general, due to randomness we can expect to find very broad resonances, though their distribution is not necessarily uniform. In the specific case of our PAuM, the bandwidth is wide enough for users to measure Raman signals with laser lines in the visible (ideally above ~500 nm) and preferentially in the aforementioned range of 650-850nm.

Reviewer 2, Q7: The position of the edge of the PAuM should be indicated in Fig. 2j, and the scan line of Fig. 2j should be marked in Fig. 2f.

Our Answer: We have changed the figure and the figure captions following the suggestion of Reviewer 2. .

Reviewer 2, Q8: The dielectric constant of the materials used in the simulation should be provided.

We state in the supplementary material in chapter S5 *Numerical Finite Element Simulations*: "For the simulations we used permittivities described in literature [7–9]."

Explicitly these are:

[7] P. B. Johnson, R. W. Christy, Phys. Rev. B 1972, 6 4370.

[8] L. Gao, F. Lemarchand, M. Lequime, Journal of the European Optical Society - Rapid publications 2013, 8, 0.

[9] C. Schinke, P. Christian Peest, J. Schmidt, R. Brendel, K. Bothe, M. R. Vogt, I. Kröger, S. Winter, A. Schirmacher, S. Lim, H. T. Nguyen, D. MacDonald, AIP Advances 2015, 5, 6 067168.

Reviewer 2, Q9: There are a few typos in the manuscript, e.g. Page 6 Fig. 1c, the unit of wavelength should be labeled. Page 7 "the energy of the plasmonic resonance

depends on one " ; Page 8 "Note the 2D-mode intensity for bare graphene and 785 nm is below the noise level" should be "Note that the 2D-mode intensity of bare graphene for 785 nm is below the noise level."? Page 10, "..the Raman scattered light for a non-porous membrane of comparable thickness, see Fig.2(c).." should be Fig.1(c)?

Our answer: We thank the reviewer for pointing out those typos. We have corrected the typos in the final version.

Reviewer #3 (Remarks to the Author):

We thank the reviewer for the evaluation of this manuscript.

Reviewer #4 (Remarks to the Author):

In this report, authors reported on the results of reducing the bulk Raman signal and increasing only the surface Raman signal by using PAuM (porous gold membrane).

As a result of measuring Raman scattering by placing the PAuM developed by the authors on a graphene/Si sample, it was confirmed that the Raman scattering on the surface was greatly enhanced.

Authors explained that the PAuM structure acts as a nano slot antenna, increasing the Raman signals from the surface and blocking Raman signals from the bulk.

Based on their experimental results, it was experimentally proven that bulk Raman scattering can be efficiently removed and only surface Raman scattering can be measured through nano slot metallic structures.

Their results are believed to suggest a new application field for porous metal materials, that is, a method to effectively separate surface and bulk Raman signals and measure only surface Raman scattering. Therefore, I expect to receive widespread attention in the fields of Raman scattering society and plasmonics field because their suggested new methodology can achieve great effect compared to the existing Raman spectroscopy.

We thank the reviewer for the favorable assessment of the manuscript.

But, I have some questions on their report, and I hope to hear from them about those.

Reviewer 4, Q1: The porous membrane presented by the authors does not seem to be able to control the polarization and desired position. In order to be useful as an actual measurement method, it seems necessary to adjust the shape and direction of the nano slot to obtain more information using incident polarization or to obtain information at a desired location on the sample. It seems good to mention the authors' alternatives methods in the revised report.

Our answer: We thank the reviewer for this question. First, we would like to point out that the random nature of the pores in our PAuM are very beneficial for multiple reasons:

1. The manufacturing of the membranes as a Raman substrate is very simple and does not require significant pre- or post-processing (apart from the evaporation process).
2. The random nature - in respect to size and aspect ratio - of the pores allows finding a suitable pore for enhancement for any given polarization- wavelength and -substrate combination.
3. Since we can fabricate the membranes at arbitrary scale, we can transfer membranes of any size (i.e. up to cm^2) on a surface of interest. This ensures that surface-sensitivity is given at any location. The Raman maps in Fig. 2 of the main paper show that we obtain some level of enhancement of the surface signal everywhere. This means that we can measure at any desired lateral position on a sample.

However, having designing PAuM with oriented (i.e. parallel) pore of specific size and aspect ratio is indeed beneficial to obtain an even better surface Raman signal. This would however require adjusting the manufacturing methods such as:

1. FIB patterning: Deposition of a continuous gold film of a desired thickness (e.g. by evaporation on an intermediate chrome layer) and subsequent patterning using a focused ion beam system. Pores of desired size, orientation and density can be readily achieved. The disadvantage of this method is the rel. slow and expensive processing.
2. E-beam lithography: Using e-beam, photoresist structures are defined on a carrier wafer where pores should be created in the membrane. Gold is subsequently evaporated on the carrier wafer, where a following lift-off process removes the predefined e-beam structures including gold on those structures. On the carrier, the gold membrane with desired pores is left behind.
3. Patterned hard mask: Patterning the carrier wafer (e-beam/RIE etching, FIB, etc.) with pores of desired aspect ratio and density yields a "hard mask" with holes. If gold is evaporated at an angle at these wafers, the holes do not/only partially fill, creating a porous gold membrane with desired structures.

Further strategies to obtain higher enhancement factors include nanoimprint lithography and block copolymers. All the methods above, however, have in common that they require a significantly higher effort both with respect to working hours processes and equipment as compared to our nanoporous membranes, which are basically fabricated in one step on wafer scale.

[Redacted]

[Redacted]

In summary, more elaborate fabrication techniques will yield higher surface Raman signal enhancement and a well-defined polarization of the light probing the sample surface, but lack the simplicity, scalability and broad-band applicability of our nanoporous membranes.

Changes to Manuscript:

Based on this question, we have added a paragraph in the discussion mentioning methods such as FIB, photolithography or nanoimprint lithography to create ordered, well-defined arrays of pores, which would enhance the Raman signal more strongly compared to PAuM at the cost of more laborious, complex and time-consuming manufacturing.

“In general, surface sensitive and bulk suppressed Raman spectroscopy is not limited to PAuM. The defining features of a plasmonic structure to enable surface sensitive Raman scattering is (i) the formation of a hotspot close to the surface, (ii) the suppression of bulk signals and (iii) the ability to sustain laser irradiation without structural damage. While other structures such as nanoparticle films or isolated bowtie structures offer one or two of these defining features, none offer the combination of all three. In contrast, PAuM and other types of plasmonic membranes fulfill these requirements. Membranes with nanopores of defined geometry, for instance, can be tailored to independently maximize bulk suppression and surface enhancement for specific excitation wavelengths. However, manufacturing this type of structures requires dedicated instrumentation (e.g. focused ion beam) often employing time-consuming serial processes, whereby one pore is created at the time, limiting the size of the membrane to a few hundreds of μm^2 . Our PAuM, on the other hand, are manufactured by simple evaporation on wafer-scale and provide broadband surface enhancement without cumbersome and laborious processing steps.”

Reviewer 4, Q2: Rather than using the porous membrane structure suggested by the authors as the above method, it seems to be more useful in various applications to directly manufacture the desired slot on a thin metal film. It seems necessary to describe whether it is possible to reduce the bulk Raman signal and obtain only the surface Raman signal.

Our answer: We agree with Reviewer 4 that manufacturing a slot directly in a metal film - see our reply to the previous question Q1 - is indeed more useful since we can tune and maximize bulk suppression via the film thickness and then directly manufacture a slot antenna of the desired properties.

This comment has shown us that we did not discuss with sufficient clarity that surface-sensitive and bulk suppressed Raman spectroscopy, as introduced by us, is not limited to the nanoporous membranes that we use in this work. In principle, any thin metallic film with nanopore-like structures that support plasmonic resonances and can be brought into direct contact with a surface of interest is suitable.

We would like to emphasize that patterning gold films is significantly more laborious than using our PAuM. Any method involving steps such as FIB, photolithography, nanoimprint lithography or any related method complicates the processing and is significantly more laborious and time-consuming. Gold membranes with pores made by these methods have their merit in specific applications and characterization where maximum enhancement is required. However, for many applications our easy-to-manufacture PAuM will provide sufficient enhancement and bulk suppression and are therefore a simple alternative to patterning metal films.

Reviewer 4, Q3: A slot that is too wide will not be able to block the bulk Raman signal and is expected to have no plasmonic field enhancement effect. Therefore, there appears to be a boundary value for the slot width at which surface enhancement/bulk depression occurs suggested by the authors. It seems necessary to present the authors' opinions on this.

Our answer: We agree with the reviewer, that there is an upper bound for the size of a pore to be plasmonically active and block bulk signals. From literature it is well known that pores with dimensions in the order of - or larger than the wavelength or the incident light will have significant transmission and no plasmonic activity.

In our PAuM there are hardly any pores longer than 300 nm, and those pores are typically randomly shaped. Light transmission without significant scattering through such a pore is strongly suppressed, even when a pore is not perfectly in resonance with a bulk Raman signal. As a coarse reference, even a tightly focussed beam shows quite some diffraction when passing a μm large pinhole. The transmission through round holes has been studied by García de Abajo Rev. Mod. Phys. 79, 1267 (2007) and characterized by means of an effective Radius R_{eff} ($R_{\text{eff}} = \text{radius}/\text{wavelength}$). The transmission rises exponentially as a function of radius and reaches about 1% for R_{eff} of 0.1. For most pores of the PAuM the transmission is very low, in the $< 1\%$ range.

Reviewer 4, Q4: The depression of the bulk signal due to the interference between SiO₂ and Si is expected to repeatedly show maximum and minimum values depending on SiO₂ film thickness. Therefore, I think that the thickness of SiO₂ thin film on substrate has optimal thickness for the suppression. I think that it's better to comment about that in the revised version.

Our answer: There is indeed an interference effect which suppresses the Si bulk Raman signal in our experiment. For a more realistic estimation of an actual bulk suppression due to the PAuM only, i.e., for a sample that features no additional layers we computed the strength of this effect. We have already discussed this effect in the supplementary material (section S4, Fig S3)). We also referred to this discussion in the main manuscript: "*Since the Si Raman signal without PAuM used as reference is affected by interference, the bulk Raman signal suppression a factor 5 to 8 higher than the experimentally obtained values, see Supporting Information S4*"

Reviewer 4, Q5: It looks good to express the lines in Figure 4 with different colors.

Our answer: We have changed the colors in Figure 4 - which is now Fig. 5 - so that they are more visually distinct.

Reviewer #5 (Remarks to the Author):

In this work, the authors report on a new technique to enhance the Raman signal from the surface of the sample selectively. Since surface-specific information is needed in many studies, this technique has a potential for an impact in the field. The manuscript is written in a clear and straightforward manner, and the presentation is mostly reasonable. There are a few points that need to be addressed before this work can be published.

We thank the reviewer for the positive assessment of the paper.

Reviewer 5, Q1: Because PAuM has to be transferred onto the surface of the sample, the contact between the sample and the PAuM should be very good. Is there any way to check this? Does it mean that this method can be used only on samples with a very smooth surface?

Our answer: The membranes indeed adhere and cover smooth surfaces very well. The exact mechanics of the membrane (adhesion to rougher surface) is a topic of further investigation. However, PAuM can be transferred to more rough substrates as well, where the only challenge could be breaking of the membrane, which would essentially limit the bulk suppression in specific spots where the underlying substrate is then exposed.

PAuM are stable enough to span over holes of several μm diameter, essentially forming a drum-like structure, without breaking. Furthermore, they are flexible on a macroscopic scale, see e.g. the transferred PAuM on polycarbonate track-etched membranes in the SI of the paper Wyss et al. ACS Applied Materials & Interfaces 14(14, 16558-16567 (2022). That membrane was transferred by heat-induced bonding and peeling the PCTE-PAuM from the support wafer.

Depending on the specifics of the roughness of the surface, the PAuM would be in contact with that surface at a specific number of positions per surface area. This corresponding contact area A_c might be described with some ratio r to the overall area A_{tot} like $r=A_c/A_{\text{tot}}$. Then the likelihood of detecting a bulk-suppressed Raman signal per illuminated and observed surface area is simply weighted with this ratio r as most of the Raman signal from the interface stem from the regions where the near field overlaps with the surface.

Reviewer 5, Q2: In the Raman spectroscopy of graphene, the SiO₂ thickness is a critical parameter. Although the value is given in Methods, it would be better to specify it when it first appears in the main text.

Our answer: We have added the value directly in the main text.

Reviewer 5, Q3: Can the authors plot the spectra in Figures 2 b-d in the unit of counts per second (CPS)? This would allow the readers to judge the actual intensity level of the signal.

Our answer: As requested, the intensity Figures 2 b-d is now given as CPS.

Reviewer 5, Q4: If the Raman signal is mostly from the top surface layer in the case of LNO on LAO, the intensity and the line shape should be independent of the thickness of the film. This is an easy check and would be strong evidence that supports the authors' claim. If the authors did not do it, it is strongly advised.

Our Answer: We thank the reviewer for this insightful question and constructive suggestion. Indeed, we appreciate the importance of verifying the independence of the Raman signal intensity and line shape from the thickness of the LNO film. In response to the advice, we grew new 7 and 14-nm-thick LNO films on LAO samples with PAuM and carried out the additional measurements. Confirming the reviewer's prediction, our results demonstrate that the spectral signatures are indeed independent of film thickness, providing strong evidence in support of our claim (for comparison of the evolution of the LNO bulk signal with the thickness, see APL Mater. 8, 061102 (2020) & PRB B 94, 014118 (2016)).

Figure: Thickness-dependent Raman measurement of LaNiO₃ thin on LaAlO₃ (a) Raman spectrum of a 14 nm LaNiO₃ thin film on LaAlO₃ using an excitation wavelength of 785 nm with PAuM and without PAuM. (b) The Raman spectrum of a 7 nm LaNiO₃ thin film on LaAlO₃ with PAuM and without PAuM using 785 nm excitation.

Changes to Manuscript:

We have included the measurements on a 7 nm and 14 nm thick film in the supplementary information and refer to this data in the main text of the paper.

Reviewer 5, Q5: The authors claim that the ratio between surface enhancement and bulk suppression can be tuned by changing the PAuM thickness. Data to support this claim are needed.

Our answer: We have already demonstrated in our previous work, that the enhancement scales with the number of pores. The number of pores scales inversely with the thickness, see image on the right from our previous work Wyss et al. ACS Applied Materials & Interfaces 14, 16558 (2022) in the Supporting information.

The light transmission through our nanoporous membranes generally follows that of a continuous film with an additional increase in transmission at the wavelengths of the nanopores' resonances, which occur around 650nm to 850nm, see Fig. 1(c) of the main paper. The figure is reproduced for convenience on the left. This clear correspondence allows us to infer that transmission is the sum of the transmission of a continuous film and the nanopores. This means a thick (thinner) nanoporous film will transmit less (more) light.

The same conclusion is drawn by Abajo (doi: 10.1088/0034-4885/17/1/302) who studied the transmission through round holes of different thicknesses. Fig.3b (log plot) in this paper shows nicely that the transmission is suppressed massively, when the thickness is increased.

Reviewer 5, Q6: According to Ref. #36, the enhancement factor for a SiO₂ thickness of 30 nm is almost 0 regardless of the wavelength. However, in Table S1 (incorrectly referred to as Table S3 in the text), the numbers substantial. The authors are advised to check the numbers.

Our answer: We thank the reviewer for pointing out this inconsistency, which needs clarification. We use 30 nm SiO₂ on Si for fabricating the PAuM. For the surface-sensitive

Raman measurements, however, we transfer graphene on 300 nm SiO₂ on Si, followed by the transfer of the PAuM. We choose 300 nm SiO₂ since this thickness provides good optical contrast to identify graphene flakes and their boundaries in order to select an appropriate area for characterization. Our enhancement factors given in table S1 are for 300nm-thick SiO₂ and agree very well with those provided in Ref. #36. We now refer to Table S1 correctly.

Changes to the Manuscript: We have now explicitly mentioned the thickness of the SiO₂/Si substrate used for Raman measurements at several positions in text as well as in the relevant figure and table captions.

Reviewer 5, Q7: In the discussion section, it is advisable to add some comments on the limitations of this technique. Some of the issues are: (1) This is essentially a destructive method. The sample cannot be used for other measurements after the PAuM is transferred. (2) Since the pore shapes and orientations are random, polarization-dependent measurements cannot be performed. (3) Since the reson is wavelength-specific, excitation-wavelength-dependent properties cannot be studied.

Our answer: We thank the reviewer for these suggestions. Regarding (1), we argue that our method is essentially not destructive. Our gold membranes can be removed easily from many materials, e.g. using a potassium-iodide (KI) gold etch if the substrate material sustains such a treatment. Furthermore, simple peeling off of the membrane can also be performed, especially for ceramics, where gold typically has a very low adhesion. Another method is simple scratching of the membrane (see image below).

In the image above, the gold membrane was found peeled after transferring (left). Measurement of the Raman signal (right) unveiled, that the 14 nm LaNiO₃ film was not damaged by the gold membrane.

Furthermore, samples with thin films or surfaces of interest are typically not entirely covered by PAuM. This is shown in the picture to the left for a 1cm x 1cm substrate that is partially covered by a PAuM. This allows for additional measurements or treatments of the pristine, uncovered surface. Raman measurements with PAuM do not require

large area coverage, and transferring i.e. 100 μm x 100 μm PAuM segments would be enough. We are currently working on methods to transfer PAuM of arbitrary size onto designated target areas of a sample.

Regarding (2) and (3), Reviewer 5 is absolutely correct in pointing out the limitations of our technique with respect to polarization and wavelength dependent Raman measurements, which exist for all spectroscopy techniques relying on plasmonic enhancement. We have also added a comment in the manuscript, see below.

Changes to the manuscript:

To highlight that PAuM removal is possible if the sample allows for it, we have added suggestions on how to remove PAuM to the supporting information S1 of the manuscript.

Furthermore, we have added a new paragraph in the Discussion section of the paper that explicitly states that the prize to pay for surface sensitivity by PAuM is that polarization and wavelength dependent Raman measurements are not available:

“The price to pay for surface sensitivity using PAuM is that the polarization and wavelength dependence of the Raman measurements are dominated by the plasmonic resonance of a nanopore. Light fields inside a nanopore are polarized primarily along the nanopore's short axis and measured Raman intensities predominantly follow the nanopores' plasmonic resonances. This means that the use of polarization and wavelength dependent Raman measurements to study the properties of a surface or thin film are rather limited. However, this limitation is not specific to PAuM since it occurs in any structure that supports plasmon-enhanced spectroscopy.”

Reviewer #1 (Remarks to the Author):

I'm glad that the revised version has solved the reviewers' concerns and the manuscript quality has been greatly improved. I suggest it can be accepted now.

Reviewer #2 (Remarks to the Author):

The authors have addressed all my concerns, and I therefore recommend the publication of the manuscript.

P.S, I suggest that the answer to the second question also be included in the supplementary, as readers who are interested in measuring the rough surface might find it helpful.

Reviewer #3 (Remarks to the Author):

Reviewer #4 (Remarks to the Author):

In the revised version, I think that authors well-answered all questions that I gave them.

Additionally, they gave us another application using their method, that is a strain measurement, and demonstrated experimentally that their method still works well.

While reading the manuscript, I raised another question on the suggested method, which is that the suggested method is a destructive or post-treated Raman measurement.

But, authors already answered about it in the rebuttal on the reviewer 5.

Therefore, I recommend that this report can be published in "Nature Communications" due to their insightful and useful development on surface-sensitive Raman measurement.

Reviewer #5 (Remarks to the Author):

The authors revised the manuscript extensively to address the comments by the reviewers. I am satisfied with the revisions and have no further comments.

Response to Reviewers 23-42440A

Reviewer #1 (Remarks to the Author):

I'm glad that the revised version has solved the reviewers' concerns and the manuscript quality has been greatly improved. I suggest it can be accepted now.

We thank the Reviewer for their positive evaluation of our manuscript.

Reviewer #2 (Remarks to the Author):

The authors have addressed all my concerns, and I therefore recommend the publication of the manuscript. P.S, I suggest that the answer to the second question also be included in the supplementary, as readers who are interested in measuring the rough surface might find it helpful.

We thank the Reviewer for their positive evaluation of our manuscript and for suggested to include our comment on probing rough surfaces. As requested, we have included our answer to question 2 in the supporting information.

Change to manuscript:

We have added a new section to the Supporting information that essentially paraphrases our discussion.

Reviewer #3 (Remarks to the Author): I co-reviewed this manuscript with one of the reviewers who provided the listed reports. This is part of the Nature Communications initiative to facilitate training in peer review and to provide appropriate recognition for Early Career Researchers who co-review manuscripts.

We hope Reviewer 3 for evaluating our work and hope that they benefited from co-reviewing.

Reviewer #4 (Remarks to the Author):

In the revised version, I think that authors well-answered all questions that I gave them. Additionally, they gave us another application using their method, that is a strain measurement, and demonstrated experimentally that their method still works well. While reading the manuscript, I raised another question on the suggested method, which is that the suggested method is a destructive or post-treated Raman measurement. But, authors already answered about it in the rebuttal on the reviewer 5. Therefore, I recommend that this report can be published in "Nature Communications" due to their insightful and useful development on surface-sensitive Raman measurement.

We thank the Reviewer for their positive evaluation of our manuscript.

Reviewer #5 (Remarks to the Author):

The authors revised the manuscript extensively to address the comments by the reviewers. I am satisfied with the revisions and have no further comments.

We thank the Reviewer for their positive evaluation of our manuscript.